# Clonal cell states link gastroesophageal junction tissues with metaplasia and cancer

Rodrigo A. Gier[1], Sydney A. Bracht[1], Jiazhen Rong [2], Raúl A. Reyes Hueros[3], Madison L. Wahlsten [1], Christopher Cote[4], Maureen DeMarshall[5], Tatiana A. Karakasheva [6], Alexandra Strauss Starling[5], Amanda B. Muir[6], Gary W. Falk[5], Nancy R. Zhang [7] & Sydney M. Shaffer [1,4] ✉

Barrett's esophagus is a common type of metaplasia and a precursor of esophageal adenocarcinoma. However, the cell states and lineage connections underlying the origin, maintenance, and progression of Barrett's esophagus have not been resolved in humans. Here, we perform single-cell lineage tracing and transcriptional profiling of patient cells isolated from metaplastic and healthy tissue. Our analysis unexpectedly reveals evidence for lineages spanning squamous esophagus, gastric cardia, and transitional basal cells at the tissue junction. We also identify lineages connecting Barrett's esophagus to both esophageal and gastric tissues. Barrett's esophagus biopsies consist of multiple distinct clones, with lineages that contain all progenitor and differentiated cell types. We discover Barrett's esophagus cell types, including tuft, ciliated, and BEST4+ cells, which we validate through both lineage relationships and spatial transcriptomics. In contrast, the precancerous dysplastic lesions show expansion from a single molecularly aberrant Barrett's esophagus clone. Together, these findings provide a single-cell view of the cell dynamics of Barrett's esophagus, linking cell states along the disease trajectory, from its origin to cancer.

Metaplasia is a response to injury in which the cell types normally found in a tissue are replaced by other, foreign cell types[1]. Because the cells of a metaplastic tissue take on a new identity, it is difficult to determine their previous identity and tissue of origin. Barrett's esophagus is a classic example of metaplasia: chronic exposure to acid reflux is believed to cause the squamous cells of the esophagus to be replaced by columnar cells resembling those of the stomach and the intestine[2]. The cell of origin for Barrett's esophagus and whether it arises through the expansion of a single cell or multiple cells remains highly debated[3]. It is also unclear whether a single progenitor cell can generate all of the cell types found in Barrett's esophagus, or if different cells contribute to separate compartments within the tissue[4]. As with other metaplastic tissues, Barrett's esophagus can undergo additional changes that lead to cancer[5]; however, the early molecular events that initiate this malignant transformation and the specific cell types involved are not fully understood[6]. Hence, to capture the origin of Barrett's esophagus and its progression to cancer, we need detailed tracking of cell

[1]Department of Bioengineering, School of Engineering and Applied Science, University of Pennsylvania, Philadelphia, PA, USA. [2]Graduate Group in Genomics and Computational Biology, Perelman School of Medicine, University of Pennsylvania, Philadelphia, PA, USA. [3]Department of Biochemistry and Molecular Biophysics, Perelman School of Medicine, University of Pennsylvania, Philadelphia, PA, USA. [4]Department of Pathology and Laboratory Medicine, Perelman School of Medicine, University of Pennsylvania, Philadelphia, PA, USA. [5]Division of Gastroenterology, Department of Medicine, Perelman School of Medicine, University of Pennsylvania, Philadelphia, PA, USA. [6]Gastrointestinal Epithelium Modeling Program, Division of Gastroenterology, Hepatology and Nutrition, Children's Hospital of Philadelphia, Philadelphia, PA, USA. [7]Department of Statistics, The Wharton School, University of Pennsylvania, Philadelphia, PA, USA. ✉e-mail: sydshaffer@gmail.com

lineages within these tissues to build a trajectory of how cells advance through the stages of disease.

As applied to the origin of Barrett's esophagus, reporter-based mouse models for lineage tracing have shown that metaplasia can develop from several different types of cells, including progenitors in the gastric cardia[7], the esophagus[8], and the gastroesophageal junction itself[9,10]. However, it is not possible to use this lineage tracing approach in humans, since it requires genetic engineering to express lineage tags. Thus, lineage tracing in human tissue has largely relied on sequencing for the detection of somatic mutations present in bulk samples, which lacks the resolution to attribute mutations to individual cells[11]. Alternatively, individual crypts or single cells can be isolated with laser capture microdissection and then sequenced for somatic mutations, but this approach is not feasible for large cell numbers[12,13]. In both of these techniques, identical mutations detected in separate samples suggest a common origin for the cells within them. However, the identity of these cells remains unresolved, making it impossible to precisely determine the cell types that share a common origin with Barrett's esophagus.

Lineage tracing of esophageal adenocarcinoma development has similarly relied on bulk sequencing of mutations in the cancer and adjacent Barrett's esophagus tissue[14–17]. In line with our understanding of cancer evolution[18], esophageal adenocarcinoma is initiated by a clonal expansion[19,20], and a recent transcriptional analysis suggests that it originates from metaplastic cells[11]. However, these bulk sequencing approaches face crucial limitations. First, bulk methods cannot determine which specific cell types within a tissue are clonally related, a fundamental limitation when studying a heterogeneous tissue like Barrett's esophagus that contains multiple specialized cell types[14–17]. Second, the cancer often shares minimal mutational overlap with the adjacent Barrett's esophagus, suggesting that bulk lineage tracing is failing to capture the small subset of cells that initially transform[16]. Third, while bulk sequencing can reveal genetic alterations, it cannot connect these changes to the specific transcriptional states. Thus, despite extensive genetic characterization of esophageal adenocarcinoma, the cell states underlying these lineage dynamics remain to be revealed.

To address these limitations from studying Barrett's esophagus with bulk analyses, the challenges are two-fold: we must have single-cell lineage tracing to identify how cells are related to each other, and we must also know the transcriptional states within these cells. To overcome these challenges, recent advances in single-cell sequencing have now enabled the detection of single-cell lineage information–in the form of mitochondrial DNA (mtDNA) mutations–with high-throughput single-cell RNA sequencing (scRNA-seq)[21,22]. Because mtDNA is passed on through cell division, each mutation labels cells that are derived from a common ancestor cell. These somatic mutations develop with age[23–25] within stem cells of epithelial tissues[26,27]. By linking cells with the same mtDNA mutations, we can reconstruct their lineages[4,13,21]. Beyond determining lineage relationships between individual cells, pairing lineage and cell state information allows us to directly measure how changes in gene expression determine cell fate.

In this study, we apply single-cell lineage tracing paired with transcriptomics to patient samples from the gastroesophageal junction, providing resolution of cellular relationships in Barrett's esophagus and its progression. By analyzing mtDNA mutations as natural lineage markers, we identify clonal relationships that span different tissue types and disease stages, allowing us to directly observe cellular connections that are otherwise undetectable in human samples. We characterize the diverse cell types within Barrett's esophagus and demonstrate that individual lineages can contain the full spectrum of differentiated cell states, including rare populations. Importantly, we reveal a striking contrast in clonal architecture between non-dysplastic Barrett's esophagus, which consists of multiple discrete lineages, and dysplastic lesions, which arise from the expansion of a single molecularly aberrant clone. This approach allows us to directly resolve the transcriptional and genetic alterations that drive malignant transformation, identifying specific molecular mechanisms that confer competitive advantages during disease progression.

## Results

### Tissues of the gastroesophageal junction can be clonally related to each other and Barrett's esophagus

The tissues spanning the gastroesophageal junction (Fig. 1a) are maintained by progenitor populations that have the potential to become dysregulated in response to chronic acid reflux[28], and thus, could be the cell of origin for Barrett's esophagus. Moreover, recent profiling of Barrett's esophagus by scRNA-seq found important transcriptional similarities between Barrett's esophagus and neighboring normal cell types from esophageal submucosal glands and the gastric cardia[11,29]. However, it is impossible to know whether single cells are indeed related from transcriptional states alone.

In order to determine whether any of these cell types native to the gastroesophageal junction were the source of Barrett's esophagus in humans, we collected a combination of pinch biopsies of the esophagus, gastric cardia, and Barrett's esophagus from 13 Barrett's esophagus patients at endoscopy and immediately subjected them to scRNA-seq and mitochondrial variant enrichment (Fig. 1b, Supplementary Table 1). Within our scRNA-seq dataset of nearly 180,000 cells, the different tissue types were readily identifiable after uniform manifold approximation and projection (UMAP) and Louvain clustering, with confirmation of expected cell types by established marker genes (Fig. 1c, Fig. S1). Mitochondrial variant enrichment of 36 of the 41 samples revealed somatic lineages unique to each tissue, as well as homoplasmic germline mutations (Fig. S2, S3). Somatic variants were identified by comparison to normal cell types, including fibroblasts, endothelial cells, and immune cells, using established methods[16,21,22]. We did not identify a relationship between variant allele frequency and clone size or cell cycle status of cells within a clone (Fig. S4). Focusing on the somatic mutations, we developed a zero-inflated beta binomial (ZIBB) model to rigorously identify cells that had acquired variants of interest whose signal was statistically higher than the predicted background noise. This model robustly identified high-quality variants across different filtering thresholds and coverage levels (Fig. S5, S6).

Throughout our analyses, we use 'lineage' to refer to cells sharing mtDNA mutations, while using 'clone' for cells that inherit both lineage markers and transformative phenotypic states such as metaplastic or dysplastic features. This framework helps determine whether the disease arose through multiple independent transformation events (polyclonal) or through inheritance from a single transformed ancestor (monoclonal) (Fig. S7). As expected, we found that the squamous esophagus and gastric cardia were polyclonal (Figs. 1e, S8), which is a well-known feature of normal healthy tissue[26]. We also did not observe a founder clone, a single dominant ancestor giving rise to the entire tissue, in any non-dysplastic Barrett's esophagus biopsies. The absence of such a founder clone suggests that non-dysplastic Barrett's esophagus can also originate from multiple clones, as supported by whole-genome sequencing studies[16,20].

To validate our mitochondrial variant-based methodology, we performed an independent analysis of single nucleotide variants (SNVs) detectable in our scRNA-seq data. We compared cells that were grouped together in our mitochondrial-based lineages with cells of the same tissue types from the same samples to assess whether genomic SNV patterns supported these groupings. Despite the sparse coverage of SNVs in scRNA-seq data[30], we found that cells sharing mitochondrial variants also shared nuclear variants at frequencies significantly higher than expected by chance (Fig. S9). This analysis showed that, when available due to sufficient sampling, genomic SNV patterns fully corroborated our mitochondrial lineage assignments.

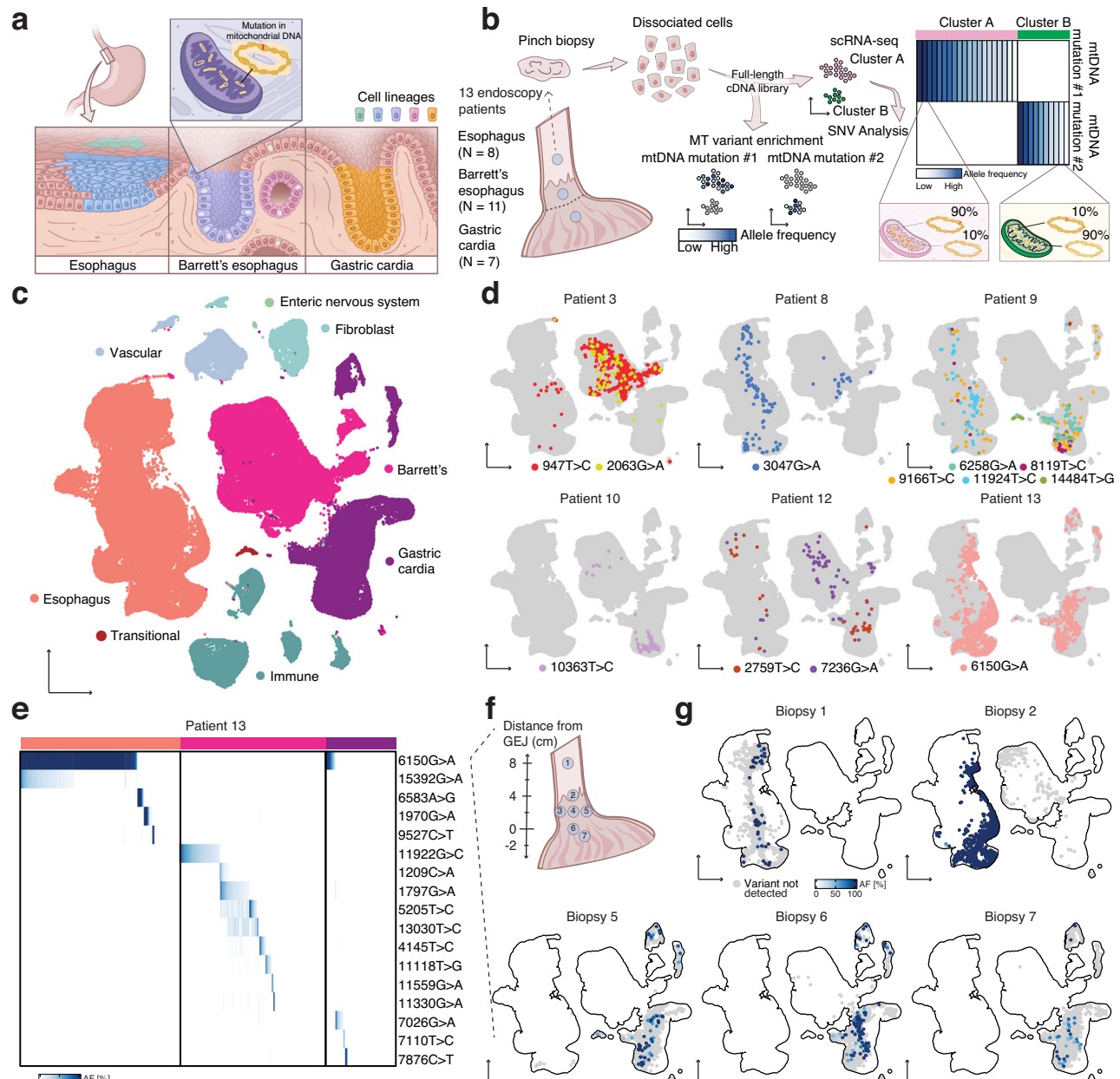

**Fig. 1 | Cell lineages in human gastroesophageal junction tissues labeled by mtDNA mutations. a** Barrett's esophagus occurs at the gastroesophageal junction between esophageal squamous and gastric cardia tissues. Clones within these tissues can be traced using distinctive mtDNA mutations. **b** Pinch biopsies of the squamous esophagus, Barrett's esophagus, and gastric cardia were collected from Barrett's esophagus patients at endoscopy and dissociated for single-cell analysis. Conventional scRNA-seq libraries can be enriched for mtDNA mutations, enabling the linking of clones to cell states. **c** UMAP of scRNA-seq of all the samples collected in this study. **d** UMAPs of scRNA-seq of all epithelial cells, where cell lineages that span multiple tissues are highlighted for individual patients. **e** Heatmap showing the allele frequencies of mtDNA mutations within squamous esophagus, Barrett's esophagus, and gastric cardia cells from all biopsies collected from Patient 13. **f** Diagram of the gastroesophageal junction region labeled with the locations of the biopsies from Patient 13. **g** UMAPs of scRNA-seq of biopsies from Patient 13 containing cells that acquired the mutation 6150G>A, plotted within the UMAP of all epithelial cells and colored with the allele frequency of 6150G>A.

In six of the patients in our study, we captured mutations in single cells that were present across tissue types (Fig. 1d), including mutations shared between squamous esophagus and Barrett's esophagus, mutations shared between squamous esophagus and gastric cardia, and mutations shared between gastric cardia and Barrett's esophagus. We found these mutations in different tissues contained within the same biopsy, as well as across separate biopsies (Supplementary Table 2). We did not observe any somatic mutations that were shared between patients (Fig. S3d), further indicating that the mutations we detected for individual patients were indeed unique lineage markers. Importantly, cells harboring these mtDNA variants maintained

expression profiles consistent with their tissue identity, suggesting that these are not due to doublet artifacts (Fig. S10). Thus, we concluded that the distinct tissues of the squamous esophagus and gastric cardia can be derived from the same clone, and that these clones can give rise to Barrett's esophagus. Notably, while these clones contain transcriptionally different cell types, the presence of a shared mutation indicates a shared somatic ancestor cell.

To investigate how lineage relationships were spatially distributed in the lower esophagus, we collected a series of at least six biopsies at measured distances from the gastroesophageal junction from two patients, 12 and 13. We captured numerous lineages across these

samples, with most restricted to a single tissue type (Figs. 1e, S8, S11); however, there were also several lineages that spanned tissues. In the case of Patient 13, we found mutation 6150G>A in five separate biopsies with cells from the squamous esophagus and gastric cardia (Fig. 1f, g); the mutation was also present in transitional basal cells that were recently shown to exist natively at the squamocolumnar junction and generate Barrett's esophagus-like tissue (Fig. 1g, Fig. S12a, b). Intriguingly, the fraction of cells that had mutation 6150 G > A was highest in biopsies close to the gastroesophageal junction and lower in biopsies farther away. In another patient with transitional basal cells containing a crossing lineage (9166T>C), we likewise observed a lower lineage fraction or failed to detect the lineage in biopsies taken away from the gastroesophageal junction (Fig. S12c, d), suggesting that the ancestor originated at or near the junction and then progeny migrated from this location.

## Barrett's esophagus consists of multiple clones with diverse cell types

To understand the role of Barrett's esophagus in the development of dysplasia, we first needed to characterize its cell types and investigate its clonal organization. In particular, we aimed to examine whether multiple cell types could be found within individual lineages in Barrett's esophagus, which would provide insight into cellular differentiation within the tissue. We collected Barrett's esophagus biopsies from eleven patients for scRNA-seq and mitochondrial variant enrichment. Of these patients, three were previously diagnosed with various degrees of dysplasia, but dysplasia was not detected in the remaining eight patients (Supplementary Table 1). When we merged the individual Barrett's esophagus transcriptomes from all the patients, including those with dysplasia, a set of consensus Barrett's esophagus cell types emerged from the overlap between known non-dysplastic samples (Fig. 2a, b).

Established Barrett's esophagus cell types, such as mature secretory (goblet and endocrine) and absorptive (enterocyte) cells, as well as OLFM4+ cells, were well represented in our data. We also captured other cell types that included tuft cells, airway-like ciliated cells, and poorly differentiated BEST4 cells that were clearly identifiable by distinct markers known from other tissues[31,32] (Figs. 2a, c, S13, Supplementary Data. 1). These specialized populations were not detected in previous single-cell characterizations of Barrett's esophagus[11], highlighting how our deeper sampling revealed additional cellular diversity within this tissue (Fig. S14). Furthermore, we consistently observed the majority of Barrett's esophagus cell types across all the biopsies, confirming their fundamental relevance to the tissue (Fig. 2b, d).

As noted earlier, we did not observe a founder clone within any of the Barrett's esophagus biopsies, which instead consisted of multiple discrete lineages that accounted for smaller subpopulations of cells (Figs. 2e, S13). Upon overlaying the lineages on our annotated UMAPs, we discovered that each lineage largely resembled the sample as a whole, both in the cell types present, as well as their relative proportion (Fig. 2f). Such remarkable differentiation potential was supported by the existence of a large pool of immature cells (Fig. S13).

While histological analysis and protein staining have been used to identify some of the cell types in Barrett's esophagus, there is currently no comprehensive mapping of the localization of these cell types based upon gene expression within the tissue. We performed spatial transcriptomics using seqFISH[33] to probe a panel of cell-type specific markers within the same tissue section that we validated with multiplexed hybridization chain reaction[34] (Figs. 3a, b, S15, S16). By integrating our scRNA-seq data[35], we mapped the cell types we identified previously onto the tissue, revealing regular expression patterns along the structure of the glands (Fig. 3c, d). We found enterocytes at the lumen, along with foveolar and goblet cells, which were also common in the middle of the gland; endocrine cells were located closer to the base of the gland, where abundant progenitor cells give rise to the more differentiated cell types above (Figs. 3b–d, S15, S16). We also directly confirmed the presence of identified cell types, including tuft (*SH2D6*), ciliated (*ZMYND10*), and BEST4 cells (Figs. 3b–d, S15, S16), and found that the ciliated cells specifically localized to the boundary of the squamous esophagus and Barrett's glands, suggesting a role in the junction of these tissues.

## A single molecularly aberrant cell can initiate the malignant transformation of Barrett's esophagus

Given that Barrett's esophagus contained multiple lineages, we next wondered whether one lineage or many lineages were represented in the transition to dysplasia. Within a biopsy collected from a patient with high-grade dysplasia, we observed two large clusters of epithelial cells by scRNA-seq (Fig. 4a); one of the clusters consisted of cells that expressed established Barrett's esophagus cell type markers, whereas the other did not, instead showing a marked loss of *PIGR* and *FAM3D*, known regulators of intestinal barrier integrity[36,37] (Fig. S17). Thus, we concluded that we had captured adjacent patches of non-dysplastic Barrett's esophagus and dysplastic tissue. In order to understand whether the cells in these molecularly distinct tissues were clonally related, we subjected them to mitochondrial variant enrichment. Consistent with our earlier lineage analysis of Barrett's esophagus, the Barrett's esophagus tissue in this sample was made up of several distinct clones, each of which accounted for a fraction of the population (Fig. 4b). Strikingly, the dysplastic tissue in this patient originated from the expansion of a single cell containing mtDNA mutation 15153G>A (Figs. 4b, S18a–c, S19), a finding further supported by concordant nuclear SNV patterns observed in single-cell RNA-seq (Fig. S9c). Our ZIBB model identified a subset of non-dysplastic Barrett's esophagus cells belonging to the same lineage as the dysplastic cells, confirming that dysplasia originated from Barrett's esophagus (Fig. S17). Within this dominant lineage, we further observed the emergence of a large dysplastic subclone that was spatially restricted within UMAP space (Fig. S18b). Such clonal expansion was not unique to the one dysplastic case; a second patient with low-grade dysplasia contained a similar precancerous outgrowth (Fig. S18d).

To further characterize the evolutionary relationships between clones in this sample, we performed phylogenetic analysis of the mitochondrial variant patterns (Fig. 4b). This revealed a hierarchical structure, with the dominant 15153 G > A mutation present across both dysplastic and a subset of non-dysplastic BE cells. An additional mutation, 5215 T > C, was restricted to specific subpopulations within this lineage. When we examined differential gene expression across these clonal populations (Figs. 4c, S18c), we identified a distinct transcriptional signature in the dysplasia origin clone, with several genes including *TPM2*, *FAU*, *PCBD1*, and *APCDD1* showing consistent upregulation. This divergence in transcriptional state observed in dysplasia compared to the well-differentiated non-dysplastic Barrett's esophagus is consistent with recent work describing phenotypic diversity in patients as a feature of progressive disease[4].

Having resolved the clonal structure of the dysplastic tissue and determined its relationship with adjacent non-dysplastic cells, we next investigated whether its transcriptional states could explain how it evolved. We noticed a large increase in the abundance of LGR5 cells, which is a known feature of dysplastic progression[38,39] (Fig. 4d). More surprising was the expression of *NOTUM*, as well as other WNT antagonists[40–43], within a subset of the dysplastic cells (Figs. 4d, S13). In situ hybridization of tissue from the same patient confirmed the scRNA-seq data and showed an expansion of the stem cell compartment (Fig. 4e). Further analysis of WNT pathway components revealed extensive dysregulation in dysplastic cells, with upregulation of WNT antagonists and a complex network of intercellular signaling that may drive competitive clonal dynamics (Fig. S18e, f). Interestingly, several genes, including *COL17A1*, which drives stem cell fitness in the skin[44], were differentially expressed in the

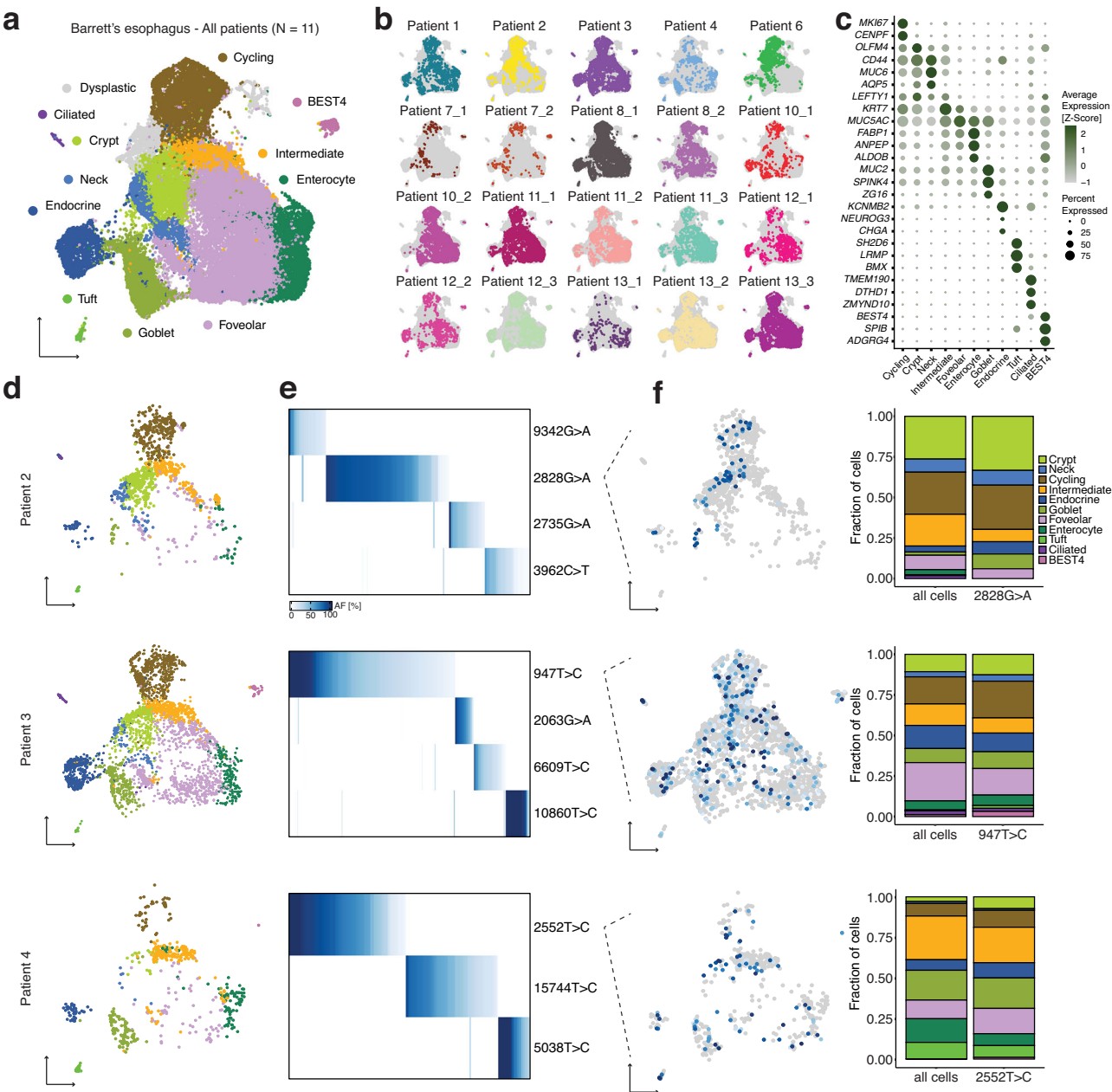

**Fig. 2 | Identity and clonal relationships of Barrett's esophagus cell types.**
**a** UMAP of scRNA-seq of all Barrett's esophagus biopsies, colored by consensus Barrett's esophagus cell types common to a majority of non-dysplastic Barrett's esophagus samples. **b** UMAP of scRNA-seq of all Barrett's esophagus biopsies, with the cells from each individual biopsy highlighted separately. **c** Bubble plot of marker genes for Barrett's esophagus cell types. **d** UMAPs generated from (b) containing only the cells corresponding to the non-dysplastic Barrett's esophagus sample indicated. **e** Heatmaps show the allele frequencies of mtDNA mutations within the Barrett's esophagus cells in the adjacent UMAP. **f** UMAP from (d) colored with the allele frequency of a representative mtDNA mutation; stacked bar graph comparing the proportion of Barrett's esophagus cell types within the representative lineage to their proportion within the entire sample.

dysplastic subclone, suggesting that new cell states that emerge within an already transformed clonal population could confer additional malignant potential (Fig. S20).

Given the level of WNT pathway dysregulation, we wondered whether there might be an underlying genetic mutation in a WNT regulator. Thus, we performed whole-exome sequencing on cells that were dissociated from the same biopsy that underwent scRNA-seq, along with matching normal squamous esophagus and gastric cardia controls (Fig. 4f). Besides confirming the presence of mitochondrial mutations in the DNA (Fig. S21), the whole-exome sequencing revealed mutations in *CDKN2A* and *TP53*, the genes most frequently implicated in dysplasia and esophageal adenocarcinoma[45,46] (Fig. 4g, h).

Additionally, as we hypothesized, there was a truncating mutation in the mutation cluster region of *APC*, a negative WNT regulator (Fig. 4h). This result raises the possibility that the dysplastic cells in this patient may have been driven by a mechanism recently discovered in pre-malignant colon adenomas, in which APC-mutant stem cells secrete NOTUM, and thereby shut down and outcompete neighboring wild-type ones[47,48].

Next, we wanted to determine whether esophageal adenocarcinoma contained Barrett's esophagus and dysplastic cell states from our scRNA-seq analysis that would further demonstrate their involvement in its development. We checked for the presence of mRNA signals[49] corresponding to the Barrett's esophagus and dysplastic cell

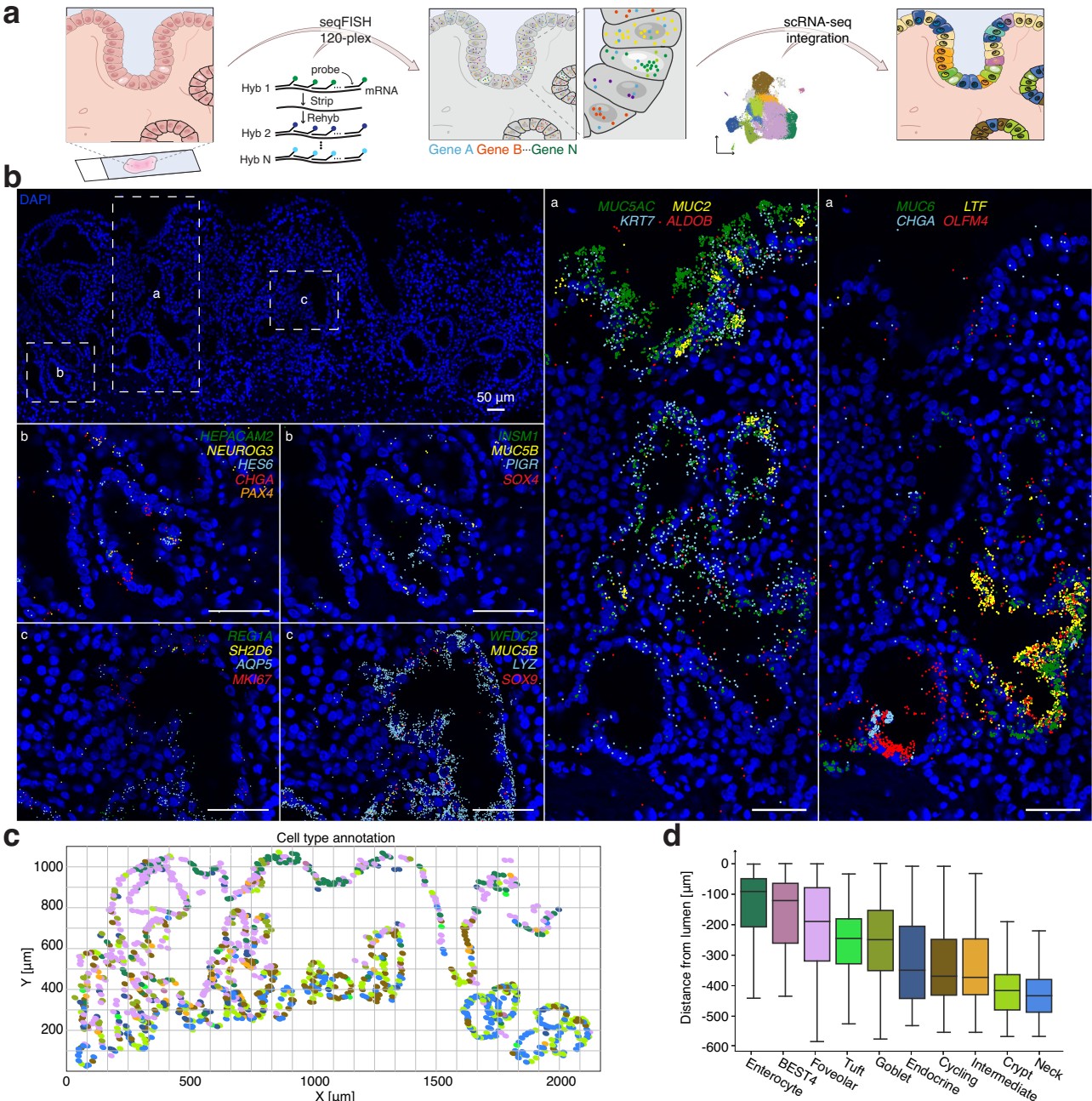

**Fig. 3 | Spatial location of Barrett's esophagus cell types. a** Schematic of the spatial transcriptomics workflow on fresh frozen sections of Barrett's esophagus tissue. scRNA-seq data was integrated with highly multiplexed RNA FISH (seqFISH) to map cell types. **b** seqFISH of fresh frozen Barrett's esophagus tissue for previously identified cell type markers from scRNA-seq (Fig. 2a, 2c); shown are a subset of marker genes. RNA Signal was called computationally and overlaid on DAPI images. DAPI staining is displayed in blue. Scale bars, 50 μm. **c** Annotation of the section from (b) with Barrett's esophagus cell type markers from scRNA-seq; the colors match the labels in Fig. 2a and Fig. 3d. **d** Boxplot of distance to the lumen for all epithelial cells in (**b**) split by cell type. Boxplot shows the median, first and third quartiles, and 1.5 times the interquartile range below the first quartile and above the third quartile; outliers were removed. Data is shown for one region of interest (ROI) from one patient. Additional patients and ROIs are in Fig. S15.

states in 88 bulk esophageal adenocarcinoma tumor transcriptomes previously analyzed by TCGA (The Cancer Genome Atlas)[46] (Fig. 4i). In 15 of the 88 esophageal adenocarcinoma tumors, we found that cell states from the dysplastic sample contributed to at least 3/10 of the bulk transcriptome (Fig. 4i). Specifically, of the three cell states that made up that dysplastic sample, the first consisted of cells with high *LGR5* and *NOTUM* expression (Fig. 4j). Therefore, in this subset of esophageal adenocarcinomas, we can conclude that there is a WNT-dysregulated transcriptional state that is similar to the cell state that

we discovered in dysplasia. In other words, Barrett's esophagus can transition to esophageal adenocarcinoma directly through the dysplastic cell state that we identified.

## Discussion

In this study, we applied lineage-resolved scRNA-seq to human samples collected from the gastroesophageal junction of patients with Barrett's esophagus and dysplasia. We identified biology at every stage of the disease. Squamous esophagus and gastric cardia cells were

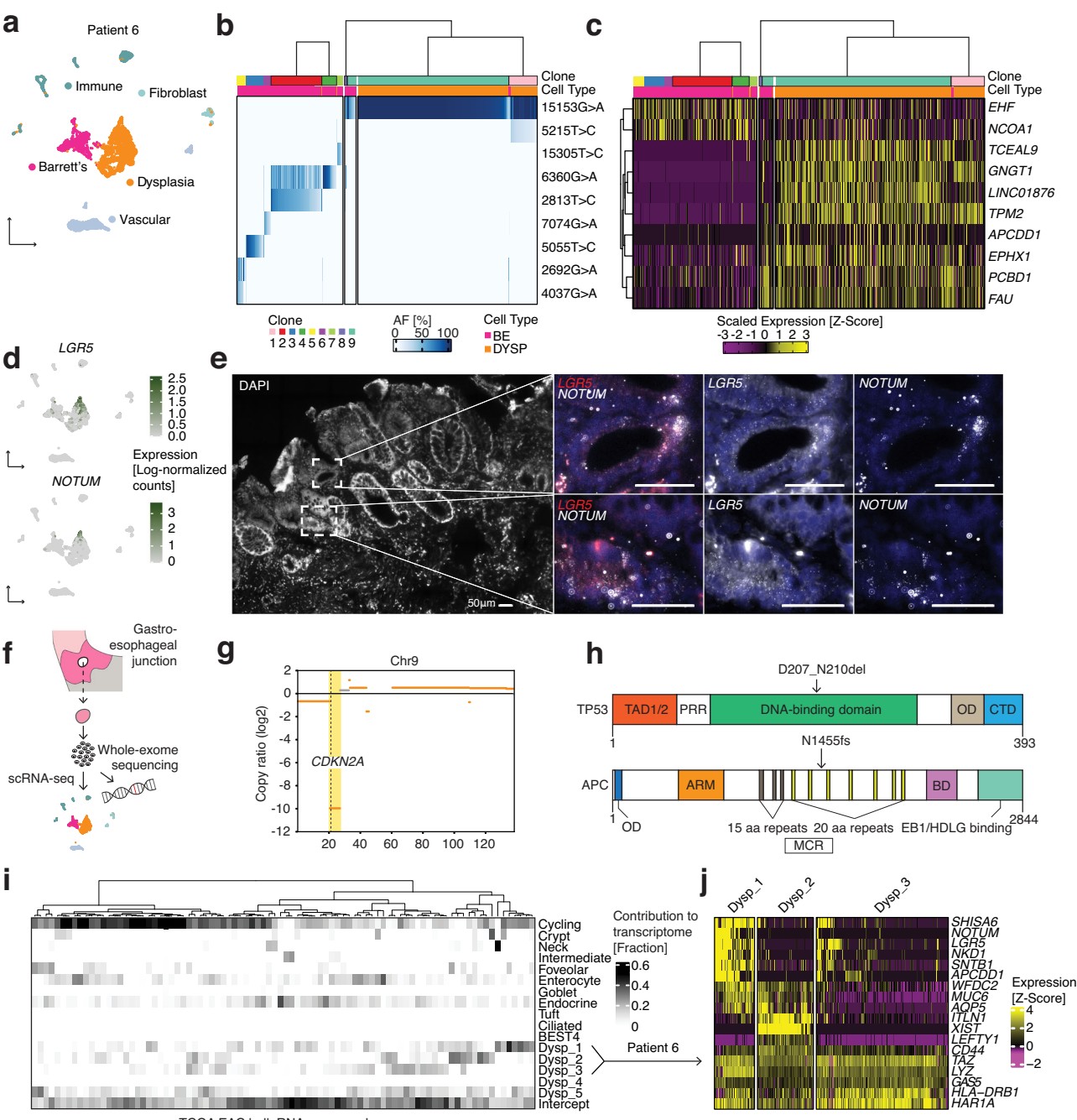

**Fig. 4 | Clonal and molecular characterization of dysplasia and its similarity to esophageal adenocarcinoma. a** UMAP of scRNA-seq of a single Barrett's esophagus biopsy from Patient 6 containing Barrett's esophagus and dysplastic cells. **b** Heatmap showing mitochondrial DNA variant allele frequencies across cells, with hierarchical clustering revealing clonal structure. Cell types are color-coded (BE = Barrett's esophagus, DYSP = dysplasia). Dysplastic cells are subsampled down to 25% of cells for visualization. **c** Gene expression heatmap of differentially expressed genes between dysplasia origin clone (containing 15153 G > A mutation) and other BE cells. **d** UMAPs from (A) featuring the expression of *LGR5* and *NOTUM*. **e** RNA FISH HCR probing for *LGR5* and *NOTUM* performed once on a fresh frozen tissue sections of a Barrett's esophagus biopsy from Patient 6. **f)** Schematic illustrating the

sequencing workflow for the cells isolated from a single Barrett's esophagus biopsy taken from Patient 6. **g** Copy number analysis from whole-exome sequencing of the same cell population from Patient 6 that underwent scRNA-seq; featured is chromosome 9, highlighting the loss of *CKDN2A*. **h** Schematics of mutations detected in TP53[63] and APC[64] proteins by whole-exome sequencing. **i** Analysis of the contribution of mRNA signals from Barrett's esophagus and dysplastic cell states to the bulk transcriptomes of esophageal adenocarcinoma tumors from TCGA. Dysp_1-Dysp_3 correspond to gene signatures specific to the dysplastic cells from Patient 6. **j** Heatmap shows genes differentially expressed within the dysplastic cells from Patient 6 grouped by clusters corresponding to the gene signatures in (i).

related to each other and Barrett's esophagus, as well as transitional basal cells found at the squamocolumnar junction. Barrett's esophagus is a polyclonal tissue in which a single cell can generate multiple mature, specialized cell types. Additionally, Barrett's esophagus can

transition to dysplasia through the expansion of a WNT-activated clone.

While scRNA-seq has proven to be an indispensable tool for mapping normal and diseased human cell states, it also has important

limitations. How these cell states are related, for example, cannot be conclusively answered without accompanying lineage information. Our results highlight the value of such information to connect cell states not just within a single phenotypic state, such as one tissue type or disease, but also between phenotypes, connecting different stages of a disease or multiple tissue types through lineage. Without lineage tracing, the scRNA-seq data alone could not have captured the relationship between transitional basal cells and esophageal squamous and gastric cardia tissues, as was missed in previous scRNA-seq work that specifically profiled the normal squamocolumnar junction[11]. Similarly, combining lineage tracing with scRNA-seq for different stages of disease allowed us to show that the transformation of Barrett's esophagus to cancer could be caused by a cascade of clonal expansions with uniquely aberrant cell states.

Our observation of multiple discrete lineages within Barrett's esophagus samples should be interpreted within the context of existing literature on Barrett's esophagus clonal evolution. While previous studies have provided evidence for clonal relationships in BE using bulk sequencing approaches[14–17], our single-cell resolution offers insights into how these lineages correspond to specific cell states. We do not claim that Barrett's esophagus originates from completely independent founders, but rather that at the resolution of our mtDNA-based lineage tracing, we observe a heterogeneous clonal structure that contrasts markedly with the dominant clonal pattern seen in dysplastic progression. This distinction in clonal dynamics between disease stages represents an important finding with potential clinical implications.

This study contributes to our broader knowledge surrounding cell plasticity at the gastroesophageal junction. We capture lineages that cross tissue types, thus relating them to each other through a shared ancestor cell. We find the highest fraction of these cells from lineages that cross the tissues of the gastroesophageal junction near the boundary, suggesting that this ancestor is near the junction. However, an important consideration is that our lineages might not contain the progenitor explicitly as the stem cell containing the mutation might not be sampled by our biopsy. Thus, the interpretation of these lineages that contain multiple tissue types is that they share an ancestor cell, not that a differentiated cell from one tissue transformed into a differentiated cell of another tissue. One possibility is that the crossing lineages in this data might arise through a multipotent progenitor at the gastroesophageal junction or gastric cardia.

While our study achieves single-cell resolution for the lineage tracing, the resolution of the lineages themselves remains a limitation. Because we are using mtDNA mutations as a marker of cell lineage, we depend upon the acquisition of mutations to provide the lineage resolution. It is thus important to consider that acquisition of mtDNA mutations is not uniform across cells, and is dependent upon the number of mtDNA per cell and the mutation rate, both of which can vary across cell types[50] and change in the setting of cancer[51]. It is also possible that the mtDNA mutations used for lineage tracing could have pathogenic effects that change their distribution in the tissue. Additionally, our approach cannot definitively rule out the possibility that some mitochondrial mutations present in founder cells may have been lost in subclones during expansion, potentially causing us to miss some lineage relationships. Such mutations could be acquired during development or with age[52–54], but then lost by a subset of cells due to mitochondrial heteroplasmy. However, our independent SNV analysis using the limited nuclear genomic information available from single-cell RNA-seq data provides supporting evidence for our findings. Despite the inherently sparse nature of SNV detection from scRNA-seq, we observed significant concordance between these genomic variants and our mitochondrial-based clonal groupings. This concordance, even with limited data, suggests that widespread mutation loss is not significantly confounding our lineage assignments. While the

orthogonal SNV validation and ZIBB provide confidence for lineage assignments, we acknowledge that artifacts are possible, particularly in the context of small numbers of cells containing a variant; thus, it is important to carefully inspect the underlying data (Fig. S5, S6, S10, and scmtVT Github) and consider more stringent filtering. In the future, we expect to see new single-cell methods for lineage tracing in tissues that will improve upon these resolution constraints.

The origins of Barrett's esophagus and its transformation to esophageal adenocarcinoma are complex phenomena. We observed evidence of clonal expansion in two dysplastic samples analyzed. In the case we examined in detail, this expansion could be traced from BE cells through a WNT-dysregulated mechanism. The generalizability of both the BE-to-dysplasia lineage relationship and the specific molecular pathways involved requires investigation in larger cohorts. In our TCGA analysis, we confirm that this cell state is relevant to esophageal adenocarcinoma, but in a subset of the cases. While this result represents a pathway for disease progression in metaplasia, it is likely one of many. Likewise, our finding that gastroesophageal junction tissues are clonally related to each other and Barrett's esophagus reconciles seemingly contradictory publications that linked Barrett's esophagus to either the squamous esophagus or gastric cardia in human samples[11,13]. However, future research will be essential to understanding how these crossing lineages develop from a single progenitor into distinct tissue types. Thus, this work further supports a growing appreciation for the fact that the paths between related phenotypic states (such as normal tissue to Barrett's esophagus, or Barrett's esophagus to cancer) will not always follow a single trajectory[55]. Studies like ours, even when they profile larger numbers of human samples, will never be exhaustive. Their value lies in providing hard evidence for the role of specific cell states in maintaining and connecting phenotypes that we could only speculate about. In the case of Barrett's esophagus, these relationships have important clinical implications that earlier methods struggled to address. This work further demonstrates the general usefulness of tracking clones across diverse cell states in disease and justifies its broader application not just at the gastroesophageal junction, but throughout the human body.

## Methods

This research complies with all relevant ethical regulations and was approved by the University of Pennsylvania Institutional Review Board (Protocol #813841), with all samples obtained following written informed consent. No statistical methods were used to predetermine sample size. The experiments were not randomized, and the investigators were not blinded to allocation during experiments and outcome assessment.

### Patient samples

Patients had a prior history of Barrett's esophagus and were undergoing routine surveillance endoscopy. The patient cohort included 14 males and 2 females, determined by self-report, with an average age of 66, sex or gender was not considered in the study design, and no sex or gender analysis was carried out as it was not deemed relevant to the biology under investigation, and because we lacked sufficient sample size. Pinch biopsies were taken from the BE tissue, as well as the normal esophagus proximal to the Barrett's esophagus segment and the gastric cardia at the discretion of the endoscopist (G.W.F). Samples were immediately transferred for processing in ice-cold Dulbecco's phosphate-buffered saline (DPBS; Corning, 21-031-CV). The Barrett's esophagus biopsy from Patient 2 was the only exception to the above; the tissue was immediately preserved in 1 mL CryoStor CS10 (Biolife Solutions, 210373) and frozen at −80 °C in an isopropyl alcohol-filled freezing container before being transferred to liquid nitrogen after 24 hours. Sample preparation required the destruction of biopsies, rendering them unavailable after processing.

## Isolation of single cells from patient tissue

Fresh pinch biopsies were washed twice in DPBS before being coarsely chopped with a scalpel and transferred into 0.5 mL DPBS solution containing 0.1 WU ml⁻¹ Liberase TH (Roche, 5401151001) and 0.5 U ml⁻¹ RQ1 DNase I (Promega, M6101). The samples were incubated at 37 °C in successive 10-minute rounds with gentle vortexing every 2 minutes and trituration every 5 minutes. At the end of each round of incubation, large tissue pieces were spun down in a microcentrifuge and the supernatant containing single cells was passed through a 70-um cell strainer fitted onto a 50-mL conical tube containing 8 mL DPBS + 1% bovine serum albumin (BSA; Sigma-Aldrich, A7906) in DPBS on ice. 0.5 mL fresh dissociation solution was added to the tissue at the beginning of each incubation step. The samples were further filtered through the 0.35 µm cell-strainer cap of a FACS tube and single cells were pelleted by centrifugation at 400 g for 4 minutes at 4 °C. Samples were washed twice in 1 mL DPBS + 0.04% BSA and finally resuspended in 50 uL DPBS + 0.04% BSA, at which point they were counted in 0.2% trypan blue (Gibco, 15250061) in preparation for 10x Genomics Chromium Next GEM Single Cell 3′ v3.1 library preparation. For the frozen BE biopsy from Patient 2, the tissue was quickly thawed at 37 °C and rinsed three times in DPBS + 10% fetal bovine serum (GE, SH30396.03) before being dissociated and filtered as described above. After the final filtering step, dead cells were removed with the Dead Cell Removal Kit (Miltenyi Biotec, 130-090-101) following the manufacturer's protocol.

## 10x Genomics Chromium Next GEM Single Cell 3′ v3.1 library preparation and sequencing

Single-cell suspensions were loaded onto a 10x Chromium Controller for GEM generation of approximately 5,000-10,000 cells for each sample. 10x Genomics Chromium Next GEM Single Cell 3′ v3.1 Dual Index library preparation was performed following the manufacturer's protocol. Library quality was confirmed on an Agilent 2100 Bioanalyzer Instrument using the High Sensitivity DNA Kit (Agilent). Libraries were paired-end sequenced on an Illumina NextSeq 550 with 28 cycles for Read 1 and 43 cycles for Read 2, as well as 10 cycles for both indices.

## 10x Genomics sequencing data mapping, and count matrix generation

Raw Illumina base call files were demultiplexed and converted into FASTQ files using CellRanger mkfastq v5.0.0. The resulting FASTQ files were loaded into STARsolo v2.7.9a for alignment to the 10x reference GRCh38-2020-A[56]. Count matrices generated with the –soloFeatures GeneFull argument were used for downstream analyses.

## scRNA-seq dimensionality reduction, clustering, and cell-type annotation

Count matrices for each sample were converted into Seurat objects in R (Seurat, v4.0.2), and genes present in fewer than three cells were removed[57]. Doublets were detected using scDblFinder (v1.8.0), with nfeatures = 3000 and includePCs = 1:20[58]. Once doublets had been filtered out, only cells with more than 300 genes and less than 30% mitochondrial reads were retained. Outlier cells containing high total counts were also removed on a sample-by-sample basis. Consistently within stomach samples, a population of cells with a large fraction of mitochondrial reads that had disproportionately high total counts was observed and confirmed to be made up of biologically important parietal cells; hence, appropriate mitochondrial and total count thresholds were set for these samples. For analyses where samples from different tissues were merged, whether within or across individual patients, count matrices were first normalized separately using NormalizeData. Merged normalized data was then processed with the standard Seurat pipeline of FindVariableFeatures, ScaleData, and RunPCA, followed by RunUMAP and FindNeighbors with 50 principal components, and FindClusters with a resolution of 0.4 or

0.6. General tissue types were identified using established high-level markers.

Batch correction of the merged Barrett's esophagus samples was performed with Harmony (v0.1.0) after subsetting for Barrett's esophagus cell types following the processing approach outlined above[59]. Two rounds of subsetting were required to remove the majority of non-Barrett's esophagus cells. Batch correction was run with the default settings, except for lambda = 14, followed by UMAP and clustering on dimensions 1:50 of the Harmony embedding and a resolution of 1.4. Barrett's esophagus cell types were identified by matching top-selected genes for each cluster found using the wilcoxauc function from the Presto package (v1.0.0) with existing annotated gastro-esophageal junction and intestinal single-cell datasets[11,31]. Certain clusters were grouped together to capture a broader cell type.

The differentiation state of Barrett's esophagus cells was determined using R package CytoTRACE (v0.3.3) with subsamplesize = 3000[60]. Inference of cell-cell communication was performed using CellChat v2[61].

## MAESTER library preparation and sequencing

Mitochondrial transcripts were enriched from the intermediate 10x cDNA libraries following the MAESTER protocol and using published MAESTER primers[22]. Briefly, previously amplified full-length cDNA was further amplified over six cycles in 12 separate PCR reactions containing primers that together spanned the mitochondrial transcriptome; 5–20 ng input cDNA was used in each reaction. PCR products for each sample were pooled and purified using Ampure XP (Beckman Coulter, A63881). Sequencing primer binding sites, adapters, and dual indices were added to the resulting cDNA over another six cycles of PCR and again purified using Ampure XP. Final MAESTER libraries were in the expected range of 2–100 ng ul⁻¹ for concentration and fragment sizes of 300–1500 bp. Libraries were sequenced on an Illumina NextSeq 550 with a v2.5 300 cycle kit, allocating 28 cycles for Read 1 and 270 cycles for Read 2, as well as eight cycles for both indices; a custom index 2 primer was used (10x-Ci5P).

## MAESTER sequencing data pre-processing and mapping

MAESTER raw sequencing data were demultiplexed and converted to FASTQ files using bcl2fastq (v2.20.0.422). Reads were trimmed of sequences that dropped below a quality threshold using a custom Python script: reads were broken up into 10-bp segments whose average quality (Q) score was calculated; the entire sequence following and including the first 10-bp segment with an average Q score lower than 25 was removed from the read. Reads that contained barcodes not present in the corresponding filtered 10x data for each sample were removed, and the remaining Read 2 FASTQ files had the library barcode, 10x cell barcode (CB), and UMI added to the read identifier. Reads were then aligned to the 10x reference genome used above with STAR[62].

## Mitochondrial genome variant calling

Mitochondrial mutations were called using maegatk, which was developed specifically for this protocol[22]. Before running maegatk, the CB and UMI were transferred from the Read 2 identifiers to the MAESTER BAM files as SAM tags, and the BAM files were merged with 10x mitochondrial reads. maegatk was run with the following arguments: -g rCRS-mb 100-mr 1. The output of maegatk included count matrices with coverage values doubled. We normalized these values when generating coverage plots in Fig. S2 and S5 to accurately represent the true per-cell coverage.

## Filtering of mitochondrial variants

Allele frequencies of mitochondrial variants in single cells were calculated in R from the maegatk output and combined with accompanying variant information, including mean coverage, mean Q score,

and allele frequency quantiles within specified cell subsets. This information was used to select for variants within each sample that had a mean coverage of >5, mean quality of >27, and AF of >25% in at least 1% of epithelial cells; such filtering identified germline and significant tissue-specific variants. The above filtering steps were based on previously published scripts (https://github.com/petervangalen/MAESTER-2021). Additionally, variants of interest were further filtered to exclude any that were found in multiple patients, since these were more likely to be artifacts.

In order to confirm somatic mitochondrial variants, we developed a zero-inflated beta binomial model to capture the background noise from contamination and technical artifacts within cells for those variants. P-values could then be calculated for the variant in question in each cell to determine whether it was improbable (and therefore true) given the modeled background noise using a zero-inflated binomial test. Finally, significant cells for a given variant were identified by calculating a false discovery rate using an empirical null distribution (based on the signal in non-epithelial cells) that accounted for differences in sequencing coverage between cells. A full description of the statistical method is provided in the supplementary information. Variants containing at least three significant cells from the ZIBB analysis after accounting for expected false positives were plotted as allele frequency heatmaps or in gene expression-derived UMAPs. Allele frequency UMAPs included cells with a minimum coverage of five for the variant of interest.

## Mitochondrial clone calling and phylogenetic reconstruction

To find related cells (clones) for phylogenetic analysis, a clustering analysis was performed on the single-cell heteroplasmy matrix generated from significant variants identified through the ZIBB pipeline using Seurat's FindNeighbors and FindClusters functions (cosine distance, k-parameter = 10, resolution = 3) with the variants acting as features as previously described[63].

Phylogenetic relationships between clusters were reconstructed using the neighbor-joining method based on cosine distances between cluster averages. Statistical support for the tree topology was assessed through a comprehensive bootstrap analysis involving 1000 replicates, in which cells within each identified cluster were resampled with replacement while maintaining cluster sizes. For each edge in the original tree, the bipartition it defined was identified by traversing the tree structure to determine the two sets of leaf nodes separated by that edge.

Support values were calculated as the frequency of each bipartition's occurrence across bootstrap replicates, with identical or reverse-ordered bipartitions considered equivalent. To calibrate these support values, a null model analysis was performed through 100 permutations, in which cells were randomly reassigned between clusters while maintaining cluster sizes. Each permuted dataset underwent the same bootstrap analysis, generating a null distribution of maximum support values expected by chance. The significance threshold was established at the 95th percentile of this null distribution. Edges were considered statistically supported only if both the edge itself and its parent node exceeded this empirically derived threshold, ensuring that significant edges represented well-supported evolutionary relationships that were unlikely to arise from the random reassignment of cells between clusters.

## Clonal differential gene expression analysis

Differential gene expression analysis was performed in two stages to identify genes associated with both dysplastic progression and clonal expansion. Following clustering of cells based on their mitochondrial variant patterns and phylogenetic relationships, cells in clusters 1, 8, and 9 were identified as belonging to a clonal lineage spanning across Barrett's esophagus and dysplastic tissue, with clusters 8 and 9 containing both Barrett's esophagus and dysplastic cells of shared clonal origin and cluster 1 representing a dysplastic subclone. Initial differential expression between dysplastic and Barrett's esophagus cells was assessed using the Seurat FindMarkers function with the Wilcoxon rank-sum test (test.use = "wilcox"), minimum detection threshold of 10% (min.pct = 0.1), and log-fold-change threshold of 0.25 (logfc.threshold = 0.25).

Due to the limited number of Barrett's esophagus cells in the identified clone, a bootstrapped differential expression approach was implemented to identify any clonal differences in gene expression. This analysis involved 100 iterations using Seurat's FindAllMarkers function (min.pct = 0.25, only.pos = TRUE), with each iteration sampling cells to match the size of the smaller group. For each bootstrap iteration, the temporary Seurat object was subsetted to contain equal numbers of cells from each group, preserving the original Seurat object structure and data normalization. Results across iterations were aggregated to calculate mean log-fold changes, mean adjusted p-values, and the frequency of significance for each gene. Genes were considered robustly differentially expressed if they achieved statistical significance (adjusted p-value < 0.05) in at least 50% of bootstrap iterations and showed consistent expression changes ( |log2FC| > 0.25). The intersection of differentially expressed genes from both the dysplasia versus Barrett's esophagus analysis and the clonal analysis was identified. The top 10 overlapping genes were selected based on their adjusted p-values from the clonal analysis bootstrap results for plotting.

## Single nucleotide variant analysis for clone call validation

To validate the mitochondrial variant-based clonal assignments, we performed an independent single-nucleotide variant (SNV) analysis using the scRNA-seq data. Putative SNVs were identified using SCReadCounts[64], a tool designed for cell-level quantification of SNV expression from scRNA-seq data. First, we used the discovery mode of SCReadCounts to identify a list of potential SNVs from each sample and patient. After compiling all samples from each patient, we then used SCReadCounts to quantify both reference and alternative allele read counts for each SNV at the single-cell level.

Following SCReadCounts, we extracted information on variant allele frequency (VAF), read depth (GoodReads), and SNV identity (SNV_ID). We filtered SNVs based on a minimum read depth threshold (≥3 reads) and required a minimum number of cells (≥4) to express each variant for reliable statistical inference. For each SNV, we then performed a comparative analysis between two groups of cells: group 1 consisted of cells belonging to a particular clone of interest (based on mitochondrial variant lineage tracing), while group 2 consisted of cells outside this clone but deliberately restricted to the same cell types and samples as group 1 to control for technical and biological confounders. This carefully matched control group design ensured that any observed differences in SNV profiles were due to clonal relationships rather than cell type-specific gene expression patterns or sample-specific effects.

We then tested for statistical significance using a two-stage approach. First, we calculated the optimal VAF threshold for each SNV using Otsu's method, which objectively separates cells with high and low variant frequencies. We then constructed 2×2 contingency tables based on this threshold and cell group assignments, and applied Fisher's exact test to assess the statistical significance of the association between variant frequency and cell group. P-values were adjusted for multiple testing using the Benjamini-Hochberg false discovery rate (FDR) method.

To assess the robustness of our findings and control for potential confounding factors, we conducted a permutation analysis with 100 iterations. In each permutation, we randomly reassigned cells to groups while maintaining the original group sizes, and repeated the statistical analysis. Permutations were stratified by sample of origin and cell type to account for potential batch effects or cell type-specific

SNV patterns. The distribution of $p$-values from permuted data served as an empirical null distribution, against which we compared the actual results to distinguish true biological signal from technical noise.

## Tissue preparation for sequential in situ RNA hybridization (seqFISH)

Pinch biopsies used for seqFISH were embedded fresh in OCT (Fisher, 23730571) and snap-frozen in an isopentane (O35514, Fisher)-dry ice bath, after which they were transferred to -80 °C for storage. 10-um tissue sections were prepared from the OCT-embedded biopsies using a cryostat (Microm) set to -20 °C and transferred to functionalized coverslips (Spatial Genomics). Tissue sections were fixed in 3.7% formaldehyde (Fisher, BP531-500) solution in PBS (Invitrogen, AM9625) for 15 minutes at room temperature, followed by 3 PBS washes of 5 minutes each. Tissue sections were permeabilized in 70% ethanol at 4 °C overnight. Following permeabilization, the coverslip was dried completely before washing with 1 mL of Clearing Solution (Spatial Genomics) for 5 minutes. The coverslip was then rinsed with 70% ethanol for 60 seconds and let air dry. Once dry, the flow cell was assembled using the functionalized coverslip and the Spatial Genomics flow cell top. 80 µL of Primary Wash Buffer (Spatial Genomics) was added to the flow cell and incubated for 5 minutes. 80 µL of primary probes were denatured at 90 °C for 3 minutes during the wash buffer incubation and subsequently added to the flow cell. The primary probe incubation took place overnight at 37 °C for 16 hours, after which the probe mix was removed and washed out with 2 washes of 80 µL of Primary Wash Buffer (Spatial Genomics). The flow cell was then incubated with 80 µL of Primary Wash Buffer for 30 minutes at 37 °C, repeating the washes and incubation a second time. Following the second incubation, the flow cell was rinsed three times with 80 µL of Rinse Buffer (Spatial Genomics) and loaded onto the Gene Positioning System (GenePS).

## seqFISH imaging and analysis

Following a preliminary scan where regions of interest were manually defined, Barrett's esophagus sections underwent 40 sequential rounds of hybridization and image acquisition for Alexa Fluor 488, Cy3B, and Alexa Fluor 647 on the GenePS[33]. DAPI was also included in the experiment. Probes for target genes were designed and synthesized by Spatial Genomics. seqFISH images were exported as.tif files. Raw seqFISH images were thresholded using the SGAnalysis software (v0.6.1). The same thresholds were used for all regions of interest on the coverslip. seqFISH images were decoded in SGAnalysis to generate a count matrix for downstream analysis.

## Integration of scRNA-seq and spatial transcriptomics

Barrett's esophagus cells were identified in the tissue based on the expression of specific canonical genes, and cells lacking this signature were excluded. Following this selection, cells with zero counts were filtered from the dataset. The remaining cells underwent normalization based on total gene counts over all genes, followed by a natural logarithm transformation and scaling to achieve zero mean and unit variance. A neighborhood graph of the cells was then computed and visualized using UMAP. Concurrently, the scRNA-seq data of Barrett's esophagus cells were aligned with the genes from the spatial genomics dataset to form a correspondence matrix. Finally, the MaxFuse pipeline was used to integrate the spatial transcriptomics and scRNA-seq datasets[35], improving the accuracy of cell type labeling. For distance measurements, the position of the lumen was determined by choosing the correct nodes from the convex hull that was found by using the Graham scan algorithm. We then calculated the distance of each cell to the line that represented the lumen.

## Tissue preparation for HCR RNA FISH

Pinch biopsies used for HCR RNA FISH were embedded fresh in OCT (Fisher, 23730571) and snap-frozen in an isopentane (O35514, Fisher)-dry ice bath, after which they were transferred to −80 °C for storage. 6–10 µm tissue sections were prepared from the OCT-embedded biopsies using a cryostat (Microm) set to −20 °C and transferred to ColorFrost Plus Microscope Slides (Fisher, 12-550-17). Tissue sections were fixed in 3.7% formaldehyde (Fisher, BP531-500) solution in PBS (Invitrogen, AM9625) for 10 minutes at room temperature, followed by 2 PBS washes of 5 minutes each. 70% ethanol was used to permeabilize cells at 4 °C overnight. Slides were stored in 2xSSC (Invitrogen, AM9765) at 4 °C.

## HCR RNA FISH

Probes for target genes were designed and synthesized by Molecular Instruments. Probe sets contained between 7 and 20 probes at a concentration of 1 µM.

Slides were removed from 2xSSC and washed once with PBS before beginning the HCR RNA FISH protocol. The protocol is a modified version of the HCR v3.0 protocol that has been previously published[34,65]. In a humidified slide chamber pre-warmed to 37 °C, 200 uL of hybridization buffer (30% formamide (Invitrogen, AM9344), 10% dextran sulfate (Fisher, BP1585-100), 9 mM citric acid (pH 6.0) (Fisher, BP339-500), 50 µg mL$^{-1}$ of heparin (Sigma-Aldrich, H5515-25KU), 1× Denhardt solution (Invitrogen, 750018), and 0.1% Tween 20 (Bio-Rad, 1610781) was added to each slide for 10 minutes. 100 uL hybridization buffer containing 0.4 pmol of each probe set was then added to the slide and covered with a glass coverslip, after which the slides were incubated for a minimum of 7 hours but up to 16 hours at 37 °C. After the probe hybridization, slides were washed three times with decreasing amounts of wash buffer (30% formamide, 9 mM citric acid (pH 6.0), 50 µg mL$^{-1}$ of heparin, and 0.1% Tween 20) and increasing amounts of 5xSSCT for 15 minutes each, followed by 2 washes in 5xSSCT alone. Samples were then pre-amplified with 200 µL amplification buffer (10% dextran sulfate and 0.1% Tween 20) for 30 minutes at room temperature. Previously ramp-cooled (0.08 °C/s) HCR hairpins (Molecular instruments) were added to 100 µL amplification buffer at a concentration of 0.6 pmol; sections were incubated in amplification solution under a glass coverslip at room temperature, concealed from light. The amplification solution was washed away with five 5-minute washes with 5xSSCT, with the last wash containing 100 ng mL$^{-1}$ of DAPI. VECTASHIELD Vibrance antifade mounting medium (Vector Laboratories, H-1700) was applied to the sample and set under a glass coverslip for at least 4 hours at room temperature before imaging.

HCR hairpins were labeled with one of the following fluorophores: Alexa Fluor 488, Alexa Fluor 546, Alexa Fluor 594, Alexa Fluor 647, and Alexa Fluor 700.

In cases where samples were re-probed for new gene targets, coverslips and mounting medium were removed by soaking the slides in PBS at room temperature overnight. Old probes were stripped using 60% formamide in 2xSSC that was applied to the slides at 37 °C for 15 minutes. Slides were then washed three times in PBS for 15 minutes also at 37 °C. After stripping, slides were wet-mounted in 2xSSC and imaged to confirm that the probes had been successfully removed. Slides were stored in 2xSSC until the next round of HCR RNA FISH, which proceeded as described above.

## Imaging of HCR RNA FISH

HCR RNA FISH samples were imaged on an inverted Nikon Ti2-E microscope with a SOLA SE U-nIR light engine (Lumencor), an ORCA-Flash 4.0 V3 sCMOS camera (Hamamatsu), a x60 Plan-Apo λ (MRD01605) objective, and filter sets 49000 ET (Chroma), 49002 ET (Chroma), 49304 ET (Chroma), 49311 ET (Chroma), 49307 ET (Chroma), and a custom set with filters ET682.5/15x and ET725/40 (Chroma). Exposure times for the hairpin dyes were between 200 ms to 1 s, while the exposure time for DAPI was 10-20 ms.

Samples that went through subsequent rounds of HCR RNA FISH were aligned using the "Align Current ND Document" (NIS-Elements AR 5.20.02) command and converted to .tif files. The resulting files were cropped and contrasted in a custom Python script that relies on the scikit-image package to perform a gamma correction operation.

## Whole-exome sequencing sample processing and library preparation

Leftover dissociated cells from the Barrett's esophagus biopsy for Patient 6 that did not go toward 10x were stored in 80% methanol, first at −20 °C for 24 hours followed by long-term storage at −80 °C; approximately 1e5 cells were saved. Total DNA was extracted from these cells following centrifugation and resuspension in 200 µL DPBS using the QIAamp DNA Mini Kit (Qiagen, 51304). 600 ng DNA were recovered, confirming our cell number estimate. Total DNA was also extracted from whole normal esophagus and gastric cardia biopsies taken from the same patient and stored in CryoStor CS10 in liquid nitrogen. Once thawed and washed in DPBS, the biopsies were chopped coarsely before being processed using the QIAamp DNA Mini Kit with the manufacturer's recommended protocol for tissue.

Whole-exome sequencing library preparation was performed using the Twist Exome 2.0 plus Comprehensive Exome Spike-in Kit (Twist Biosciences, 105036) and sequenced on an Illumina NovaSeq 6000 at the Center for Applied Genomics at the Children's Hospital of Philadelphia with at least 100x coverage.

## Whole-exome sequencing data preprocessing, somatic variant calling, and copy number analysis

Sequencing data was preprocessed following the Genome Analysis Toolkit's (GATK) Best Practices[66]. Briefly, FASTQ files were converted to unmapped BAMs and checked for Illumina adapter sequences. Raw reads were aligned to the GRCh38 reference human genome with the Burrows-Wheeler Alignment Tool's Maximal Exact Match algorithm (v0.7.10), after which duplicates were marked. Unless otherwise stated, the above were performed using Picard (v1.141).

Preprocessed whole-exome sequencing data were analyzed with Mutect2 (GATK, v4.2.5.0) to identify short somatic mutations including single-nucleotide polymorphisms and insertions and deletions. The Barrett's esophagus sample was run with the two matched normals, as well as the publicly available germline resource somatic-hg38_af-only-gnomad.hg38.vcf.gz (GATK). Somatic variant calls were filtered using FilterMutectCalls (GATK) and loaded into the Integrative Genomics Viewer (v2.12.3) for identification of high-quality variants in known esophageal adenocarcinoma regulators[45,46].

Copy number alterations in the Barrett's esophagus sample were estimated using CNVkit (v0.9.9) with the default run settings[67,68]; a pooled reference was generated from the normal esophagus and stomach samples. Copy number results were plotted directly within CNVkit using the function scatter.

## Barrett's esophagus single-cell reference analysis of bulk transcriptomes

Count matrices of bulk transcriptomes were downloaded from the TCGA-ESCA project (https://portal.gdc.cancer.gov/projects/TCGA-ESCA). The Python package cellSignalAnalysis identified reference signals derived from the Barrett's esophagus scRNA-seq analysis in the bulk transcriptomes, while accounting for the possibility that the mapping was incomplete[49]. Reference signals were generated from clustered scRNA-seq data by summing the raw counts for each gene across all the cells in each cluster, after which the summed counts were normalized to sum to one. cellSignalAnalysis was run using Seurat clusters that, in some cases, were subsequently grouped to define broader cell types. The contribution of these subclusters was combined by cell type in the output.

## Reporting summary

Further information on research design is available in the Nature Portfolio Reporting Summary linked to this article.

## Data availability

The single-cell RNA raw sequencing data are deposited in the Gene Expression Omnibus (GEO) under accession number GSE291080. The spatial transcriptomics data are deposited in GEO under accession number GSE288235. The MAESTER data are deposited in GEO under accession number GSE287855. The previously published data compared in S14 from Nowicki et al. (2021) are available from the European Genome-phenome Archive (EGA) EGAD00001005438. All raw microscope images are available on Zenodo under accession 17229323. The whole exome sequencing data generated in this study have been deposited in the database of Genotypes and Phenotypes (dbGaP) under accession number phs003949 [https://www.ncbi.nlm.nih.gov/projects/gap/cgi-bin/study.cgi?study_id=phs003949.v1.p1]. These data are available under restricted access due to patient privacy regulations and the presence of potentially identifying genetic information. To request access, researchers should submit a Data Access Request through the dbGaP Authorized Access system. Access requests must include a Data Use Certification signed by the investigator and institutional signing official, outlining the proposed research use and data security measures. The NIH Data Access Committee typically reviews requests within 2–4 weeks of submission. Once approved, access is granted for one year with the possibility of annual renewal, and data will remain available in dbGaP according to NIH data sharing policies.

## Code availability

Code for the ZIBB model is available under an Apache License 2.0 on Zenodo under accession 17229078[69] [https://doi.org/10.5281/zenodo.17229078] and on Github [https://github.com/sydshaffer/scmtVT/tree/v1.1]. The Apache License 2.0 permits unrestricted use, modification, and distribution of the code for commercial and non-commercial purposes, with the requirement that the copyright and license notices are preserved. Diagnostic plots from ZIBB and FDR threshold analysis outputs are also available on the same accession. The code used for making figures has been deposited at Zenodo under a Creative Commons Attributions 4.0 International License under accession 17185171[70] [https://doi.org/10.5281/zenodo.17185171]. The Creative Commons Attribution 4.0 International license permits unrestricted use, modification, and distribution of the code for commercial and non-commercial purposes as long as appropriate credit to the original authors and indication of any changes made is given. Our code used to process MAESTER samples was modified from Peter van Galen's MAESTER protocol[22] [https://github.com/petervangalen/MAESTER-2021], which is licensed under the MIT Copyright License © 2021 vangalenlab. The copyright notice and license information have been retained in our relevant source files in the Zenodo repository 17185171.

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

## Acknowledgements

We thank the patients who contributed to this study. We thank all members of the Shaffer Lab for their valuable feedback and support. We thank L. Bugaj, P. Cámara, A. Raj, and K. Zaret for insightful comments on the manuscript. We thank C. Ren and D. Dykes for assistance with illustrations. We thank SciStories for their work illustrating Fig. 1a. We thank L. Dolinsky, G. Park, and A. Benítez for assistance with sample retrieval. We thank R. Wu and A. Sahasrabuddhe, as well as the Genomics Facility at the Wistar Institute, for assistance with sequencing. We thank the Center for Applied Genomics at the Children's Hospital of Philadelphia for assistance with whole-exome library preparation and sequencing. We thank K. Frieda and Spatial Genomics for technical support with seqFISH. R.A.R.H. acknowledges support from the NSF Graduate Research Fellowship (DGE-1845298). A.B.M. acknowledges support from NIH grant R01DK124266. G.W.F. acknowledges support from the Center for Molecular Studies in Digestive and Liver Diseases (NIH/NIDDK P30 DK050306). S.M.S. acknowledges support from the NIH Director's Early Independence Award DP5OD028144, the Institute for Regenerative Medicine at the University of Pennsylvania from the Larry and Mickey Magid award, the Institute for Translational Medicine and Therapeutics of the Perelman School of Medicine at the University of Pennsylvania (NIH NCATS UL1TR001878), and the American Cancer Society (RSG-23-1152597-01-CDP). S.M.S. is the Bakewell Foundation Innovator of the Damon Runyon Cancer Research Foundation (DRR-81-24). S.M.S., A.B.M., G.W.F., N.R.Z., and A.S. acknowledge support from NIH NIDDK (R01DK135729). S.M.S. and N.R.Z. acknowledge support from NIH NIGMS (R01GM149671).

## Author contributions

Conceptualization, R.A.G. and S.M.S.; Methodology, R.A.G. and S.M.S.; Investigation, R.A.G., R.A.R.H., and S.A.B.; Software, R.A.G., R.A.R.H., and J.R.; Formal Analysis, R.A.G., S.A.B., J.R., M.L.W., N.R.Z., and S.M.S.; Data Curation, R.A.G. and R.A.R.H.; Writing – Original Draft, R.A.G and S.M.S.; Writing – Review and Editing, R.A.G., S.A.B., M.L.W., C.J.C., and S.M.S.; Visualization, R.A.G., S.A.B., R.A.R.H., M.L.W., and S.M.S.; Resources, M.D., T.A.K., A.S.S., A.B.M, and G.W.F; Funding Acquisition, A.B.M., and S.M.S.; Supervision, N.R.Z. and S.M.S.

## Competing interests

The authors declare no competing interests.
