## [Peer Review File · Nature Communications]

Clonal cell states link gastroesophageal junction tissues with metaplasia and cancer

Corresponding Author: Dr Sydney Shaffer

Version 0:

Reviewer comments:

Reviewer #1

(Remarks to the Author)

In this study, Gier and colleagues performed single-cell transcriptomic analyses with concurrent lineage tracing from mitochondrial DNA (mtDNA) mutations on primary tissue samples from 13 patients with Barrett's Esophagus (BE). These data enabled the authors to map different cell types at the gastroesophageal junction (including BE) and partially determine the lineage relationship between these cells. The authors claimed that these data "elucidate the full trajectory of Barrett's esophagus disease" (line 84). To this reviewer, this claim is unjustified: Most of the findings are confirmatory and do not strengthen, refine, or challenge the current understanding of the evolution of BE and its progression to EAC. This reviewer also finds the lineage analysis require further validation/refinement. Overall, major revisions are needed before the manuscript can be considered for publication at Nature Communications.

Below are the specific critiques.

A. Technical

I start with the technical critique because this is directly related to the conceptual critique (B). The central issue here is what inferences can be drawn from the mtDNA mutations.

The first question is how to interpret the different allelic frequency distributions of mtDNA mutations. For example, in Fig. 1E, the first mutant appears to have 100% or 0% AF, whereas the second appears to have a wider AF range. What does it mean when different mutant alleles (1797, 5205, 13030) in the same subclonal population have variable AF's? I expect the AF to correlate with the copy of the mutant allele and the AF distribution to be informative about the timing of the mutant. But to use mutant AFs for lineage tracing requires a quantitative model of AF distribution as mitochondrial DNA is replicated or segregated during somatic cell division. I could not find such information.

The second question is how to interpret the pattern of subclones with different (largely mutually exclusive) mutations and use this information for lineage tracing. It is reasonable to group cells with a specific mutation into a subclone; however, this does not rule out the possibility that some of these subclonal mutations were in fact present in the founder cell, but lost in subclones during clonal expansion. Therefore, I am unsure how to draw inferences about the clonal/cell type relationship of BE and adjacent cells (e.g., Fig.1E, S4, S5).

Given the importance of lineage tracing in this study, the clarification of the above questions is essential for evaluating the validity of the biological conclusions. The authors should show all the cells (including those without any detectable mtDNA mutation) in figures like Fig.1E with a phylogenetic tree of different subclones resolved by subclonal mutations. The authors may want to group cells both by cell type (as in Fig.1E) and by mutations (as in Fig. 2F). When a subclone contains different cell types, some statistical assessment is necessary to exclude misclassification (either of subclonal identity of cell type).

B. Conceptual

It is unclear what "gaps" (line 72) the authors attempted to bridge and what new insight was generated in this study.

The authors brought up several open questions about Barrett's Esophagus (BE) and Esophageal Adenocarcinoma (EAC).

These include: a. The cell-of-origin and clonal composition of BE tissue (line 42); b. Whether a single cell can generate different cell types in BE (line 43-44); c. What molecular events initiate the transition from BE to EAC (line 45-46).

The process of somatic cell expansion implies that all BE cells must have descended from a single common ancestor. If BE cells show different transcriptional/cell states (Nowicki-Osuch et al., Science 2021), such diversity must have arisen from their common ancestor. Thus the observation that a single cell produces subclones of different cell types (Fig.1D) is unsurprising. The author commented that there was no founder clone in any of the BE biopsies (line 117). This conclusion drawn based on the absence of ancestral mtDNA mutations shared by all BE cells is flawed because the absence of detectable mtDNA mutations shared by all BE cells does not count as evidence for the absence of an ancestral founder cell (see Technical critique). In other words, the presence of multiple BE subclones does not represent any evidence against the possibility that all these subclones descended from a single ancestor cell.

The claim that BE cells do not all descend from a single founder clone (i.e., polyclonal origin of BE tissue) may be supported if there are different subclones (by mtDNA mutations) that each consist of both BE cells and non-BE cells. I could not find definitive evidence of this. In Fig.1E, all mtDNA mutations in BE cells (magenta) are exclusive to BE. The same pattern appears to hold in Patient 3, 4 (with a minor band in the gastric cardia group for one mutation), 8, and 12 (with one mutation present in multiple cell types shown in Fig. S5B). Without further clarification of the interpretation of MAF in these cells, I am not convinced about the result in Fig. 5B. Moreover, the rarity of different cell types including BE sharing a single ancestral mtDNA mutation suggests that BE cells in most of these patients descended from an ancestor that is different from the ancestors of all other cell types. This supports the single clonal origin of BE, instead of suggesting "that non-dysplastic Barrett's esophagus also originates from multiple clones" (line 118).

Moreover, the analyses of BE and EAC genome evolution by a variety of approaches (single cell, multi-focal, longitudinal) in prior studies have provided extensive evidence supporting a single clonal origin of BEs as well as revealing genetic changes during the BE-to-EAC transition (see Refs. 14,15,16,18,19,20). Again, the divergence between BE/EAC or between different BE subclones does not represent evidence for a polyclonal origin of BE cells.

The only remaining open question is what is the cell type of the most recent common ancestor of BE clones. However, this question cannot be addressed by the analysis of BE cells, because the emergence of a macroscopic BE clone inevitably prevents the analysis of their ancestor.

In summary, the author's analysis provided no new information about BE evolution, the cell type of the ancestor cell(s) of BE clones, or the evolution of BE cells and/or its progression to EAC (the current patient cohort does not include EAC patients). The only potential novel finding from the application of concurrent lineage tracing and transcriptome analyses of single cells (line 84-85) is about the different cell types in BE and the surrounding tissue (esophagus, gastric cardia). To this reviewer, almost all of the authors' claimed findings in line 84-91 are either overstated or overinterpreted.

C. Minor critiques

1. The classification/annotation of different cell types in BE (Figure 2A-D) appears to be the most important data of the paper. The authors should compare the marker genes (Fig. 2C) or gene sets with results from prior studies (e.g., Nowicki-Osuch et al.) for validation.
2. Figure 3 primarily illustrates the spatial location of cell types within the tissue, which is not directly relevant to the main topic of this manuscript -- addressing the origin and clonality of Barrett's esophagus. Therefore, it would be appropriate to move Figure 3 to the Supplementary data.
3. Regarding Fig. 4A, what's the reference for using the loss of PIGR and FAM3D as markers for dysplasia? Regarding Fig. 5D, how confident is the assignment of BE cells with the same 15153 G>A mutation as in dysplasia?
4. What are the cell types of BE cells that share the same mutation as dysplasia?
5. The identification of APC mutation in one sample is insufficient for the conclusion that BE progression to EAC is "driven by a mechanism recently discovered in premalignant colon adenomas" (line 226).

(Remarks on code availability)

Reviewer #2

(Remarks to the Author)

I co-reviewed this manuscript with one of the reviewers who provided the listed reports as part of the Nature Communications initiative to facilitate training in peer review and appropriate recognition for co-reviewers.

(Remarks on code availability)

Reviewer #3

(Remarks to the Author)

The manuscript presented by Gier et al aims to identify clonal relationships between individual cells of Barrett's Esophagus (BE) and their relationship to Dysplastic BE and normal tissues surrounding BE. The majority of the author's conclusions are derived from two technologies: 10x genomics scRNA-seq and mitochondrial lineage tracing based on the heteroplasmy derived from scRNA-seq data. Firstly, the authors observe that the transitional cells residing at the squamocolumnar junction share mitochondrial mutations with normal gastric and esophageal epithelia suggesting their common origin. Next, they focus on BE and they show that BE is a polyclonal tissue with multiple lineages supported by different mitochondrial variants. A similar analysis was used to show that all cell types of BE arise from a single population of stem/progenitor cells. A different pattern was observed in dysplastic BE, wherein, in 2 patients, a single founder variant was observed. Finally, the authors observe the dysregulation of WNT signalling as a main driver of the progression of BE from non-dysplastic to dysplastic states.

This study is a technically well-executed scRNA-seq work with good analysis and the application of mitochondrial lineage tracing is a good addition to this work. The addition of spatial analysis further strengthens the authors' discoveries of cell populations present in BE samples and significantly improves on previous work in the BE field. Finally, the development of the ZIBB model (with limitations listed below) and its application to all samples is another significant contribution. The authors state that their data support the hypothesis that Barrett's Esophagus (BE) is of polyclonal origin and dysplasia originates from a single founder cell. However, it cannot be said that the presence of multiple clones derived from heteroplasmic variants supports the hypothesis that BE arose as a polyclonal structure. Equally valid is a hypothesis that BE arose as a clonal structure and heteroplasmic mutations occurred after BE development. Since this study contributes to two significant debates that exist in the field of BE and upper-GI biology, namely the question of BE origin and BE clonality status and its role in disease progression, extra care is needed in the application of exciting, new technologies. As a result, additional scrutiny of the following issues is warranted:

1. The authors claim that the absence of evidence to support a clonal origin of BE is sufficient evidence of polyclonal origin. This conclusion has weak reasoning (absence of evidence is NOT evidence of absence), cannot be strongly supported by the data, and is difficult to reconcile with the existing literature including literature cited by the authors (references 16, 20, 26). In particular, Paulson et al. find both polyclonal and clonal origin of BE can be supported by their WGS (depending on the patient), although in all cases, the authors found evidence of shared mutations within individual cases. If Gier and colleagues aim to support the polyclonal origin of BE, they ought to investigate the likelihood of such an outcome in their data. Indeed, in all figures, the authors did not indicate how many BE or normal epithelial cells do not have any heteroplasmic events (in cells with sufficient coverage). For example, BE cells in Patient 13 (figure 1E) do not share heteroplasmic events with any of the adjacent normal epithelia suggesting that it either A) originated from multiple clones that had heteroplasmic events prior to BE development (as suggested by authors) or B) originated from a single clone that did not have any of the detected heteroplasmic events (suggesting that normal tissue of origin also did not have them) and these heteroplasmic events arose later in the natural history of the disease or C) a mixture of (A) and (B). Since the authors did not mention anywhere if all cells with sufficient coverage in variant regions are plotted on heatmaps, the choice between (A), (B) and (C) cannot be made. A rudimentary (ignoring confounding of selection and the genetic drift of heteroplasmic events) approach to this problem would estimate the proportion of cells (cells with minimal coverage in all variants of interest should only analyzed) carrying heteroplasmic events in all tissues. This can then be used to estimate the likelihood that the observed events in BE are of polyclonal origin (the proportion of all cells with heteroplasmic events is higher than in the matched normal epithelium) or clonal (the events are less frequent and arose later in the history of the diseases). The current iteration of the manuscript can only support a notion that BE is polyclonal in its states not in its origin.

2. Using data from three patients (9, 12 and 13) the authors find that some mitochondrial mutations are shared between squamous, gastric and transitional and an additional three patients share mutations with BE epithelia. They then draw a conclusion that due to shared mutations; their progenitor originates at the junction (lines 143-144). Firstly, the definition of "progenitor" needs to be clearer. In line 137 transitional basal cells are called progenitors, and in the next section (lines 146 onward) progenitor cells are stem-like cells of BE. Do authors suggest that transitional cells are stem-like cells of gastric cardia and squamous epithelium, or do they suggest that these three (four if BE is included) epithelial types are of common embryonic origin when heteroplasmic mutations spanning normal epithelial could have arisen (e.g. PMID: 28238948)? More importantly in multiple cases (e.g. 8119T>C in Patient 9 (squamous), 7236G>A in Patient 12 (squamous tissue), 947T>C in Patient 3 (squamous tissue)) the number of cells that support the conclusions drawn is very low. Probably only mutation 6150G>A in patient 13 and the existence of multiple mutations in patient 9 provides sufficient support to the authors' conclusions. As a result, careful examination of the following is required:

a. In the case of mutations that are only shared within a single biopsy (e.g. 947T>C) it is imperative that the authors check in the tissues with the low number of cells supporting the conclusion (e.g. squamous in the case of 947T>C) that the cells carrying mutations are not doublets. I understand that automated tools were used for doublet detection, but they are not perfect in a complex setting and manual investigation would be critical.

b. Secondly, the identification of the significant variants relies on the ZIBB model presented in the supplementary text and GitHub repository. In his model, multiple parameters are derived from the distribution of variants observed in the control "normal" samples. However, it is of concern to me that the distribution of coverage in the "normal" cells often does not follow the distribution of other cell types (highlighted in figure S11, "normal" cells tend to have lower total coverage). Did the authors investigate if the variables derived from one distribution of cells translate into another? Secondly, the methods do not explain what filtering of reads has been done prior to coverage and vaf calculation (line 409). Since the mean quality (it is not clear if it of read or base, line 409) is over 27, it is likely that lower-quality data will be fed into the model. Accounting for base calling and alignment quality is standard in somatic variant calling (e.g. mutect2). The authors should clarify if the results stand when allele counting prior to ZIBB model execution is performed using only high-quality bases (e.g. uniquely mapped reads, with base calling quality above 30 (or any other empirically derived quality threshold of bases)).

3. The seqFISH data presented in Figure 3 and S9 are of high quality and beautifully highlight the distribution of cells in the individual glands. The quantification presented in figure C and D are also excellent. However, other slides and quantification in Figure S8 seem to follow a different distribution of cells to the one seen in the main figure. In particular data from Patient 16 seem to have Enterocyte cells (dark green) at the top and bottom of the slide. It is likely that this slide is a cross-section or

tissue has been folded. An H&E image would be helpful in the proper identification of tissue orientation.

4. For clarity, it would be helpful if all heatmaps, in addition to tissue type, also had information about biopsy ID. This would help with the interpretation of figures.

Minor:

1. Figure 1C, ENS needs explaining (only explained in the related supplementary figure)

(Remarks on code availability)

The code is reproducible, example data is shared and the code can be installed.

Reviewer #4

(Remarks to the Author)

The study investigates Barrett's esophagus and its progression to esophageal adenocarcinoma through the use of single cell RNA-seq and mtDNA-based lineage tracing. By analyzing patient samples from the gastroesophageal junction, the researchers identify clonal cell states and track their evolution from healthy tissue through metaplasia to cancer. The study provides insights into the cellular dynamics and heterogeneity underlying these conditions. Key findings include the identification of common lineages between squamous esophagus and gastric cardia cells, the existence of multiple clones within Barrett's esophagus arising from a single progenitor cell, and the transition of Barrett's esophagus to esophageal adenocarcinoma through the expansion of a single molecularly aberrant clone.

1. The validity of this study relies heavily on the accuracy and comprehensiveness of the mtDNA variants identified to achieve high-resolution and fine-scale clonal structures. The study acknowledges that the limited number of captured mitochondria restricts the accuracy and resolution of lineage tracing. It may be worth conducting some sort of sensitivity analysis by sub-sampling reads and testing out different variant selection methods beyond the author's combination of mgatk and scmtVT.

2. The inference that there must be a single Barrett's esophagus progenitor cell based on the fact that each lineage contained most of the cell types within Barrett's esophagus appears to lack sufficient evidence. Other possibilities, such as the presence of multiple progenitor cells or a hierarchical differentiation process, cannot be ruled out based on this observation alone.

3. The authors concluded that dysplasia in Patient 6 originated from Barrett's esophagus because they found that non-dysplastic Barrett's esophagus cells belong to the same lineage as the dysplastic cells that contain the mtDNA mutation 15153G>A. However, in the heatmap of Figure 4B, the 15153G>A mutation did not appear in Barrett's cells. This observation raises questions about the authors' conclusion that dysplasia originated from Barrett's esophagus. Further investigation and analysis are needed to clarify this discrepancy and determine the true origin of dysplasia in Patient 6.

4. A cell-state-aware phylogenetic tree can provide a comprehensive understanding of the origin and relationships among different lineages in Barrett's esophagus with dysplasia. It can help unravel the links between clonal cell states and the development of dysplasia and cancer.

5 The authors propose a mechanism in which Barrett's esophagus transforms through a single clone with a WNT-dysregulated cell state. While the study provides valuable evidence for the role of specific cell states in maintaining and connecting phenotypes in Barrett's esophagus, I believe the conclusions are primarily based on expression analysis and would benefit from further validation. Moreover, additional analyses, such as examining receptor/ligand interactions related to WNT signaling, could provide more insight into the underlying mechanisms. By incorporating the validation and additional analyses, the authors could strengthen their conclusions and provide a more comprehensive understanding of the cellular dynamics involved in Barrett's esophagus.

6. May I assume all relevant data will be deposited in a standard FAIR-compatible public repository such as NCBI SRA if the paper is accepted for publication? I do not think a personal google drive link is appropriate.

(Remarks on code availability)

The R package 'scmtVT' is provided in a GitHub page. This is reasonable.

The data and processed data are available in a personal google drive. There is almost no systematic metadata associated with the data. It is very difficult for me to navigate through the many layers of directories. This way of storing and sharing data is not suitable, and not FAIR-compatible.

Version 1:

Reviewer comments:

Reviewer #1

(Remarks to the Author)

In this revised manuscript, the authors have carried out various new analyses in response to my comments. In addition to presenting results from these analyses, the authors also discussed the strengths and limitations of their analytical approach.

I commend the authors for the rigor of their work and presentation. The data and analytical results provide strong support for the main conclusion of the paper. Although some of the issues raised in my original review are not fully resolved, I understand that these are beyond the technical capabilities of the experimental methods employed in this study. I therefore recommend publication of the revised manuscript without additional data/analyses. I have two suggestions for text/presentation revisions that the authors may consider.

1. The definition of a 'clone' is critical for readers to appreciate the authors claim that BEs contain multiple clones whereas dysplastic lesions are derived from a single molecular aberrant BE clone. A simple definition of a clone as a cell population derived from a single ancestor cell is inadequate because both BE's and dysplastic BEs share a common ancestor and therefore can be considered 'clonal.' An alternative definition of clones based on shared mutations is also inadequate because any somatic tissue is polyclonal due to the continuous acquisition of somatic mutations in each cell division. Based on my understanding, a clone considered by the authors refers to a cell population that share/inherit a certain phenotype (e.g., transcriptional state) from their ancestor. Thus, a BE is considered polyclonal if there exist multiple BE clones each arising from and carrying the phenotypes of a BE ancestor, but the ancestors of different BE clones do NOT inherit a common phenotype of their ancestor. By contrast, dysplastic BE is monoclonal because their ancestor is expected to have already acquired the phenotype of all the dysplastic BE cells. It may be helpful if the authors can draw a schematic diagram of somatic clonal expansion and phenotypic variation to highlight these two different scenarios. It will also strengthen the conclusion of the paper if the authors can describe the criteria for defining a founder clone (either BE clone or dysplastic BE clone).

2. Given that the authors only analyzed dysplastic BE from one patient, they should soften their conclusions derived from this single sample. These include the monoclonal origin of dysplastic BE's (there is always a chance that microscopic dysplastic lesions were not adequately captured in the biopsy), and the inference that APC inactivation promotes dysplastic BE progression.

2.

(Remarks on code availability)

Reviewer #2

(Remarks to the Author)

(Remarks on code availability)

Reviewer #3

(Remarks to the Author)

The authors have substantially modified their study and introduced significant changes to the manuscript including new analysis that support their model conclusions. Importantly the authors have clarified central message of this study. This version of manuscript provides important evidence in support of polyclonal nature of non-dysplastic BE and clonal nature of dysplastic BE. Secondly, their scRNA-seq data from over 100K cells are a great additional to the field, further strengthened by the spatial validation of these results. In general, the authors have done a good job modifying this manuscript to retain information that are of importance to the field. I believe that the investigation of the cellular composition of BE and the clonal-polyclonal dichotomy between BE with dysplasia and nondysplastic BE should be the central messages of this study. In the case relatedness of tissues of the upper gastro-intestinal tract, I still have few concerns that stem from potential issues with the procedure for the discovery of somatic mitochondrial variants. At this point I would like to raise two points:

1) In the response to my earlier comments, the authors have provided commentary on the mechanistic of empirical FPR calculation using binning methods every 20 coverage counts. However, how do authors calculate FDR for coverage bins which do not have any cells in the control condition. E.g. on figure S6, variant 11891G>A does not have any cells in the control condition with coverage >200 but there are many cells with this coverage in the BE samples. How was FDR calculated for those bins?

a. it would be helpful if such plots were readily available for each sample either as a single supplementary pdf or at least as fixed link supplementary data (not google drive folder).

2) Secondly, the choice of empirical FDR is commendably, however, my concern is that the procedure for FDR calculation might be too relaxed. As a result, I have concerns whether some of the mutations identified in specific samples are not true discoveries but false positive results not accounted for by the authors statistical procedure.

For example, mutations 2063G>A in stomach of patient 3 is supported only by 3 or 4 cells (figure 1d), however closer examination of the underlying data (https://drive.google.com/file/d/1LQlgsJtd5YZs7bGjoZfA2_IDzXeeLbUb/view) demonstrate that the significant cells have low total count of alleles indicating potential false positive result not detected by the authors' FDR calculation. This stands in stark contrast with the data for the same allele in the BE samples (https://drive.google.com/file/d/1GMQfJe95Q9dv2b4I6vMq9IU1Cs_0RgJj/view?usp=drive_link). Here, the mutated alleles are plentiful and they are supported by cells with both high and low total count of reads.

To reiterate, I do not think the statistical procedures used in the ZIBB model are incorrect. My concern is that the chosen FDR calculation procedure is not stringent enough to support authors conclusions about shared lineage between tissue types when the number of cells supporting this conclusion is limited.

a. As a result, it would be important that at minimum the authors introduce additional discussion points to account for the fact that some of the shared mutations might be artefacts of their analysis. Alternatively, the authors should investigate if the conclusions stand if more stringent FDR procedures are applied.

b. Also, the abstract still contains a statement "Our analysis unexpectedly revealed that the squamous esophagus and gastric cardia contained cells belonging to common lineages that also included transitional basal cells; both esophageal and gastric tissues were also related to Barrett's esophagus". In light of my concern, this statement cannot be easily supported by the results presented in this study and it should not treat it as the main conclusion of their analysis that should be included in the abstract.

Minor point:

In figure S7, the authors introduced heatmaps "Ordered by Biopsy". I believe that unlike figures to the right of these heatmaps, all cells were plotted. As a result, there is a discrepancy between the panels - e.g. mutations 2063G>A does not exist in blue (esophagus) Sample on the left panel but man cells in the same sample have it at low AF on the right panel.

(Remarks on code availability)

The code is accessible, well annotated and appears to be reproducible.

Reviewer #5

(Remarks to the Author)

This revised manuscript by Gier et al focuses on the study of Barrett's esophagus, a premalignant condition of esophageal adenocarcinoma. The major claims of the article are that:

1. Squamous esophagus and gastric cardia contain cells belonging to common lineages that included transitional basal cells
2. Normal esophageal and gastric tissues contain clones that are related to Barrett's dysplastic cells (which suggests a common ancestral origin, or possible sources from both organs as the cell of origin)
3. Barrett's esophagus in its early state contains multiple clones (possibly multiple origins)
4. More dysplastic cells contain a single molecularly aberrant clone (derived from one of the BE clones/subclones in a classic clonal evolution process)

Evidence to support these claims mostly comes from a series of biopsies from various patients that were analyzed through single-cell RNAseq and a modified version of MAESTER to enrich and profile mitochondrial mutations for lineage tracing. Methods seem to rely on state-of-the-art, validated techniques, for the most part (the ZIBB model is a nice addition), and the quality controls and stringency measures seem to be appropriate and well demonstrated in supplementary figures. I could also find and run the ZIBB code from github.

Fig 1 presents the general evidence for claims 1-2. I found support figures S8 (SNV validation) and S9 (doublet/cell-state testing) very convincing. Fig 2 presents the evidence that Barrett's contains multiple clones and each clone appears to be multilineage, being in basically every cell type (claim 3). Fig 3 presents spatial transcriptomics of Barrett's, which validates some new BE cell types, which is really good. Fig 4 presents the evidence of claim 4, that dysplastic cells are basically a single clone derived from one of the BE clones (subclones?). WGS indicates the likely mutational process, and the evidence of transcriptionally similar cells in at least some esophageal adenocarcinomas (main supporting evidence that these dysplasias identified would go on to become EAC).

Regarding the points raised by other reviewers, Reviewer 1 technical comments are valid, and highlight some of the known limitations of mt-based tracing. First, mitochondrial tracing is rarely based on fully homoplasmic mutations... usually there is some heteroplasmy, and the distributions of VAFs are hard to interpret due to limitations (copy numbers, dropouts, etc.). I think the reply by the authors is fair, confounders are excluded as much as possible, and they even went to the extent of validating variant detection and calls using the ZIBB model and performing SNV-based validation (Fig S8). I would be satisfied with this answer.

Regarding Reviewer 1 conceptual comments, I think that it is also valid to argue that all these cells have an ancestral clone. The whole animal is coming from a single ancestral clone if we consider the zygote. With our current knowledge of how long it takes for mitochondrial variants to arise and stabilize at high heteroplasmy, and the fact that these clones differentiate to most BE cell types, it is reasonable to argue that BE may have independent origins, even if (as the Reviewer points out) the killer experiment is to show normal non-BE cells with the same mutations in order to exclude the fact that these mt variants arose within the BE-tissue. In any case, I believe there is significant novelty in showing that BE is a polyclonal ecosystem. Meaning, at the very least, the authors are proving the existence of multiple BE "stem cells" and their fate behaviors, which is super interesting and novel. The manuscript could better highlight in the abstract the finding and validation of novel BE cell types (validation both from the clonal perspective, since they share mt variants with other BE types, and from the seqFISH experiment). Finally, I do agree with their MINOR comment 5, regarding some overstatements.

Reviewer 3 point 1 goes very much along the lines of Reviewer 1 at least regarding concepts. I think the authors have done a good job at toning down their polyclonal origin model in favor of focusing on a polyclonal state model. I also want to highlight that the replies regarding QC of variants (Fig S5) and doublets (Fig S9) are easy to follow and understand, and I agree generally with their interpretation.

I also want to comment on the Reviewer 4 point 2 regarding the lack of sufficient evidence for a single progenitor cell for BE. This is a classical problem that emerges with mt-based tracing, since we cannot fully time the emergence of the mitochondrial variants. In one patient, the stabilized variant may be late and fate-restricted. In the next patient, the variant may be even representative of clonal origins pre-BE. This is why it is so important in this kind of study to present these data cautiously without overinterpreting. But this is not to say the study is not novel (as reviewer 1 so adamantly states). I think it pushes the boundaries of what has been done before, and it provides guidance for future endeavors trying to solve these questions (with mt mutations or otherwise).

In Figure 4b/c, the tree connection between BE and Dysplasia cells is very cool, but really hard for me to see. It's too small, and there's too few cells in the BE lineage block of the heatmap. I suggest downsampling the Dysp cells for plotting the heatmap (they're all mostly the same anyways!).

In summary, from my assessment at the current review stage, considering the manuscript in its current state, I believe it would still benefit from some further text adjustments, but no further experiments or analysis would be necessary.

(Remarks on code availability)

I think the code and data should be deposited somewhere more definitive with a proper DOI before final acceptance.

Version 2:

Reviewer comments:

Reviewer #3

(Remarks to the Author)

I would like to thank the authors for thorough investigation of potential issues with the selection of empirical FDR. The textual changes applied as sufficient for readers to consider potential issues and I do not have further comments.

(Remarks on code availability)

Zenodo repository contains all information needed for the reconstruction of individual parts of analysis. github repository (<https://github.com/jiazhen-rong/scmtVT>) contains all code need to rerun the analysis using ZIBB model.

Reviewer #1

(Remarks to the Author): Expert in cancer genomics and evolution, single-cell sequencing, computational and statistical genomics, and Barrett's Esophagus

In this study, Gier and colleagues performed single-cell transcriptomic analyses with concurrent lineage tracing from mitochondrial DNA (mtDNA) mutations on primary tissue samples from 13 patients with Barrett's Esophagus (BE). These data enabled the authors to map different cell types at the gastroesophageal junction (including BE) and partially determine the lineage relationship between these cells. The authors claimed that these data "elucidate the full trajectory of Barrett's esophagus disease" (line 84). To this reviewer, this claim is unjustified: Most of the findings are confirmatory and do not strengthen, refine, or challenge the current understanding of the evolution of BE and its progression to EAC. This reviewer also finds the lineage analysis require further validation/refinement. Overall, major revisions are needed before the manuscript can be considered for publication at Nature Communications.

Below are the specific critiques.

A. Technical

I start with the technical critique because this is directly related to the conceptual critique (B). The central issue here is what inferences can be drawn from the mtDNA mutations.

The first question is how to interpret the different allelic frequency distributions of mtDNA mutations. For example, in Fig. 1E, the first mutant appears to have 100% or 0% AF, whereas the second appears to have a wider AF range. What does it mean when different mutant alleles (1797, 5205, 13030) in the same subclonal population have variable AF's? I expect the AF to correlate with the copy of the mutant allele and the AF distribution to be informative about the timing of the mutant. But to use mutant AFs for lineage tracing requires a quantitative model of AF distribution as mitochondrial DNA is replicated or segregated during somatic cell division. I could not find such information.

We appreciate the reviewer's insightful observations about the variable allele frequencies (AFs) of mtDNA mutations. Indeed, we observe different frequencies across mutations (as seen in **Fig. 1e** with variants like 1797, 5205, and 13030). This heteroplasmy is a well-documented biological phenomenon in mitochondrial genetics. In our analysis pipeline, variants that pass through our ZIBB model and are designated as clones typically show a characteristic pattern: most cells have AFs close to zero, while cells within a clone have AFs that surpass our statistical threshold. Our ZIBB model is intentionally conservative, with highest statistical power for detecting variants at high frequency, allowing us to confidently identify cells within a clone using a rigorous approach.

Following the reviewer's suggestion, we conducted additional analyses to investigate potential correlations between AF and various parameters. We examined the distribution of AFs across the three tissue types (**Fig. S4, also below in Response Fig. 1**), finding variable distributions in each tissue. We also tested for relationships between AF and clone size or proliferative state (using G2M quantification in Seurat), but found no significant trends at either the clone level or single-cell level. This lack of correlation is not surprising, as cells initially derived from highly proliferative progenitors may now exist as fully differentiated cells, obscuring direct relationships between current proliferation states and historical clonal dynamics.

Response Fig. 1, Fig. S4. Analysis of mitochondrial allele frequencies across tissue types and cell proliferation states. (a) Boxplots showing the allele frequency distributions of significant mtDNA variants detected in squamous esophagus (pink, left), Barrett's Esophagus (magenta, middle), and gastric cardia (purple, right). Each variant is identified by position and nucleotide change, with the number of significant cells shown in parentheses. (b) Scatterplot displaying the relationship between median allele frequency and number of cells containing each variant. Points are colored by tissue type. (c) Scatterplot showing the relationship between median allele frequency and percentage of G2M-high cells (proliferative cells) within each variant lineage. (d) Single-cell allele frequency (y-axis) versus G2M score (x-axis) plots for individual variants across all tissues, demonstrating the lack of correlation between mitochondrial variant allele frequency and cell proliferation state.

To determine the timing of mutations across tissues, future studies could analyze the distribution of shared versus private mutations as has been done in the literature (Passman et al. 2023) and others. However, this approach would require more extensive sampling and deeper sequencing than our current methodology provides. The technique we employed in this study does not detect a sufficient number of mutations with the necessary confidence level to make reliable timing estimates. Additionally, our analytical workflow deliberately filters out private mutations that appear in only a few cells, further limiting our ability to construct accurate mutation chronologies. A modified approach with greater sequencing depth and broader sampling would be required to establish the temporal sequence of mutations across the different tissues at which point the allele frequency could be better utilized to examine the timing of mutations.

The second question is how to interpret the pattern of subclones with different (largely mutually exclusive) mutations and use this information for lineage tracing. It is reasonable to group cells with a specific mutation into a subclone; however, this does not rule out the possibility that some of these subclonal mutations were in

fact present in the founder cell, but lost in subclones during clonal expansion. Therefore, I am unsure how to draw inferences about the clonal/cell type relationship of BE and adjacent cells (e.g., Fig.1E, S4, S5).

The reviewer raises an important methodological concern about mtDNA-based lineage tracing. Indeed, mitochondrial heteroplasmy could theoretically lead to mutation loss in some subclones, confounding lineage relationships. To address this fundamental issue, we developed an orthogonal validation approach using nuclear DNA variants.

Our independent SNV analysis provides a critical test of our lineage assignments. Unlike mitochondrial DNA, nuclear DNA is not subject to heteroplasmy effects, making it ideal for validating lineage relationships. We systematically compared the nuclear SNV patterns between cells assigned to the same mitochondrial-defined clone versus matched control cells (same cell types from the same samples). Using Otsu's thresholding method followed by Fisher's exact tests with FDR correction, we found significantly more shared nuclear variants within mitochondrial-defined clones than expected by chance.

As shown in **Fig. S8** (included below as **Response Fig. 2**), cells within the same mitochondrial-defined clones display a strong enrichment of low p-values for shared nuclear variants compared to permuted controls. This statistical signal would be absent if mitochondrial mutations were being substantially lost during clonal expansion, as the nuclear variants would not show concordant patterns. The clear distinction between real and permuted data distributions directly addresses the reviewer's concern and provides strong evidence that our lineage assignments represent true biological relationships rather than artifacts of mitochondrial heteroplasmy.

Response Fig. 2, Fig. S8. Validation of mitochondrial variant-based lineage assignments using RNA detected single nucleotide variants (SNVs). (a) Schematic workflow of the SNV analysis pipeline. (b) Density histograms showing distributions of p -values from Fisher's exact tests for real data versus permuted data across multiple patients and mitochondrial variants. Enrichment of low p -values in real data indicates significant associations between mitochondrial variants and RNA detected SNV patterns. (c) Variant allele frequency distributions for representative SNVs in Patient 3, showing significant differences between cells within mitochondrial lineages (teal) and other cells (orange). (d) VAF distributions for top RNA-detected SNVs in Patient 6 (15153 lineage), demonstrating concordant SNV patterns that support mitochondrial variant-based lineage assignments.

The concordance between SNV patterns and our mitochondrial-based clonal groupings provides orthogonal confirmation of our lineage assignments. If substantial mutation loss were occurring (such as all cells originally having mutations that were subsequently lost, creating the observed subclone pattern), we would not see significant signal over permutation in our SNV analysis.

While we acknowledge that loss of heteroplasmy could still occur in a subset of cells, potentially causing us to miss some cells that should be part of a clone, our statistical approach suggests these effects are minimal. We have added this limitation to the discussion section of the manuscript in paragraph 3 as shown below:

“Additionally, our approach cannot definitively rule out the possibility that some mitochondrial mutations present in founder cells may have been lost in subclones during expansion, potentially causing us to miss some lineage relationships. Such mutations could be acquired during development or with age (McCracken and Wells 2017), but then lost by a subset of cells due to mitochondrial heteroplasmy. However, our independent SNV analysis from nuclear genomic DNA (which is not subject to the same heteroplasmy effects as mtDNA) suggests these effects are minimal, as we observe significant concordance between genomic SNV patterns and mitochondrial-based clonal groupings that would not be expected if mutation loss were widespread.”

Given the importance of lineage tracing in this study, the clarification of the above questions is essential for evaluating the validity of the biological conclusions. The authors should show all the cells (including those without any detectable mtDNA mutation) in figures like Fig.1E with a phylogenetic tree of different subclones resolved by subclonal mutations. The authors may want to group cells both by cell type (as in Fig.1E) and by mutations (as in Fig. 2F). When a subclone contains different cell types, some statistical assessment is necessary to exclude misclassification (either of subclonal identity or cell type).

We appreciate the reviewer's thoughtful suggestions for strengthening our lineage analysis presentation. We have revised our figures to provide a more comprehensive view of clonal relationships. We have updated the visualization in **Fig. 1e** to show cells that do not have detectable mitochondrial mutations. We have also included heatmaps for all patient samples showing this information in **Fig. S7**. The cells are visualized both when grouped by cell type and when regrouped by mutation status, allowing direct comparison of these two important perspectives.

Response Fig. 3, Part of Fig. S7. Visualization of mitochondrial variant distributions for 2 patients (all are included in Fig. S7). Heatmaps showing mitochondrial variant allele frequencies in single cells from Patients 3 and 4, with cells ordered by cell type or by biopsy sample. Cell types are indicated by colors at the top of each heatmap (BE = Barrett's esophagus, SQ = squamous esophagus, GC = gastric cardia, IM = immune, FB = fibroblast, VS = vascular, NS = not specified). Sample origins are color-coded (es = esophagus, be = Barrett's esophagus, st = stomach). For clarity, only 2 representative patients are shown here; comprehensive heatmaps for all patients are included in Supplementary Figure S7.

While the lineage information in Fig. 1 is relatively straightforward with largely independent mutations, the dysplastic case in Fig. 4 reveals a more complex mutational hierarchy. To address this complexity, we have added a new phylogenetic analysis for this case in **Fig. 4b,c and Response Fig. 4** which integrates the phylogenetic relationships with gene expression patterns, providing insight into how transcriptional changes correlate with clonal evolution.

Response Fig. 4, Main Fig. 4-c. Phylogenetic analysis of dysplastic lineage in Patient 6. The figure shows hierarchical clustering and phylogenetic relationships of cells based on mitochondrial variant patterns. (a) UMAP of scRNA-seq of a single Barrett's esophagus biopsy from Patient 6 containing Barrett's esophagus and dysplastic cells. Below are UMAPs of the epithelial cells with the color scale depicting allele frequency for mitochondrial variants 15153G>A and 5215T>C detected in Patient 6. (b) Heatmap of mitochondrial variant allele frequencies with hierarchical clustering identifying nine distinct clonal groups (numbered 1-9, color-coded at bottom). (c) Gene expression heatmap of differentially expressed genes between the dysplasia origin clone and other Barrett's esophagus cells, showing distinct transcriptional patterns that align with clonal groupings. The lineage annotation (color bar at top) highlights the dysplasia origin clone (red) compared to other Barrett's esophagus clones (green).

To address concerns about potential misclassification, we have performed additional analyses of our cell type assignments and added a new **Fig. S9**. We first examined cell type markers for squamous cells (*KRT4*, *KRT5*), gastric cells (*GKN1*, *LIPF*), and Barrett's esophagus (*TFF3*, *REG4*) and plotted the distribution of expression values for these genes in the cells with and without the variants that cross tissue types. In each instance, the cells followed the expected expression pattern for the cell type that they were classified as in the analysis (**Fig. S9a**). We also visualized total UMI counts in these cells to determine whether there was an unexpectedly high or low level of UMIs detected as such might point to other sources of artifact (doublets or empty droplets). Indeed, we found that the cells with these variants had UMI levels that were equivalent to the distribution of the cells that do not carry the variant (**Response Fig. 5, Fig. S9b**).

Response Fig. 5, Fig. S9. Validation that cells containing cross-tissue mitochondrial variants are not doublet artifacts. (a) Violin plots showing expression levels of canonical marker genes for squamous epithelium (KRT4, KRT5), gastric cardia (GKN1, LIPF), and Barrett's esophagus (TFF3, REG4) in cells with specific mitochondrial variants (947T>C and 2065G>A in Patient 3; 2759T>C and 7236G>A in Patient 12) compared to non-variant cells of the same tissue type. Top row shows Barrett's esophagus cells, middle row shows gastric cells, and bottom row shows squamous cells. Cell numbers for each group are indicated below each plot. **(b)** Distribution of total UMI counts in cells containing the same mitochondrial variants as in (a), demonstrating that variant-containing cells have UMI distributions comparable to non-variant cells of the same tissue type, rather than the abnormally high UMI counts typically observed in cell doublets. Statistical significance was assessed using two-sided Wilcoxon rank-sum tests with Benjamini-Hochberg correction for multiple comparisons (n.s. = not significant, $p > 0.05$; * $p < 0.05$; ** $p < 0.01$; *** $p < 0.001$).

B. Conceptual

It is unclear what "gaps" (line 72) the authors attempted to bridge and what new insight was generated in this study.

We acknowledge the reviewers' concerns regarding our characterization of Barrett's esophagus as polyclonal. We have revised our discussion to better situate our findings within the existing literature. Previous whole-genome sequencing studies by Ross-Innes et al. (2015) and Paulson et al. (2022) have demonstrated clonal heterogeneity within Barrett's esophagus, suggesting multiple clonal origins rather than a single founder clone. Our single-cell approach adds nuance to these findings by revealing the gene expression profiles of the lineages within Barrett's esophagus (Fig. 2f).

In contrast to the clonal heterogeneity observed in non-dysplastic Barrett's esophagus, our data clearly show that dysplastic lesions arise from the expansion of a single molecularly aberrant clone (Fig. 4b, Fig. S17). In one patient, this dysplastic clone exhibits distinct transcriptional features, particularly WNT pathway dysregulation. This contrast between the clonal architecture of non-dysplastic and dysplastic Barrett's esophagus provides important insights into the progression of the disease. We have modified our language throughout the manuscript to more precisely describe these observations, avoiding overstated claims about polyclonality while highlighting the cellular and molecular differences between non-dysplastic and dysplastic tissue that our single-cell lineage approach uniquely reveals.

The authors brought up several open questions about Barrett's Esophagus (BE) and Esophageal Adenocarcinoma (EAC). These include: a. The cell-of-origin and clonal composition of BE tissue (line 42); b. Whether a single cell can generate different cell types in BE (line 43-44); c. What molecular events initiate the transition from BE to EAC (line 45-46).

The process of somatic cell expansion implies that all BE cells must have descended from a single common ancestor. If BE cells show different transcriptional/cell states (Nowicki-Osuch et al., Science 2021), such diversity must have arisen from their common ancestor. Thus the observation that a single cell produces subclones of different cell types (Fig.1D) is unsurprising. The author commented that there was no founder clone in any of the BE biopsies (line 117). This conclusion drawn based on the absence of ancestral mtDNA mutations shared by all BE cells is flawed because the absence of detectable mtDNA mutations shared by all BE cells does not count as evidence for the absence of an ancestral founder cell (see Technical critique). In other words, the presence of multiple BE subclones does not represent any evidence against the possibility that all these subclones descended from a single ancestor cell.

We appreciate the reviewer's thoughtful critique regarding the interpretation of clonal structure in Barrett's esophagus. See further discussion above regarding relevant literature in the field and how our work builds on this. We agree that the absence of detectable mtDNA mutations shared by all BE cells does not conclusively rule out a single ancestral founder cell. However, our data provides multiple lines of evidence that challenge a simple single-origin model.

First, if BE originated exclusively from cells in neighboring squamous esophageal or gastric cardia tissue, we would expect to observe the preservation of pre-existing mutations from these tissues in BE. In our dataset, 42% of squamous esophageal cells and 38% of gastric cardia cells contain detectable mutations. Yet across our cohort of 13 patients, we never observe the widespread preservation of such founder mutations in BE tissue, a notable contrast to our dysplastic samples where we do observe clear founder effects.

Second, we observe specific clonal patterns that are difficult to reconcile with a simple founder clone model. For example, in Patient 3, we identify two different non-overlapping clones (947T>C and 2064G>C) that both contain cells within the BE cluster. This pattern would not be expected if a single clone had given rise to the entire BE tissue. Similarly, with Patient 12, we observe a clone (7236G>C) that contains many BE cells alongside squamous and gastric cells, suggesting a more complex developmental relationship than unidirectional differentiation from a single BE progenitor.

To address concerns about potential misclassification, we have performed additional analyses confirming that these cell type assignments are accurate based on canonical marker expression patterns and UMI distributions (**Fig. S9, Response Fig. 5 above**).

We have revised the text to clarify these findings while acknowledging the limitations of mtDNA-based lineage tracing for definitively proving or disproving a single founder origin. The presence of multiple discrete lineages within BE is consistent with previous whole-genome sequencing studies (Ross-Innes et al., 2015; Paulson et al., 2022), though we recognize that earlier common ancestors without detectable mutations cannot be ruled out.

The claim that BE cells do not all descend from a single founder clone (i.e., polyclonal origin of BE tissue) may be supported if there are different subclones (by mtDNA mutations) that each consist of both BE cells and non-BE cells. I could not find definitive evidence of this. In Fig.1E, all mtDNA mutations in BE cells (magenta) are exclusive to BE. The same pattern appears to hold in Patient 3, 4 (with a minor band in the gastric cardia group for one mutation), 8, and 12 (with one mutation present in multiple cell types shown in Fig. S5B). Without further clarification of the interpretation of MAF in these cells, I am not convinced about the result in Fig. S5B.

We agree that the majority of the mutations observed in BE cells are independent from the neighboring esophagus or stomach samples, which is consistent with tissue-specific clonal expansion. However, we do find rare but significant instances where mutations exist across different tissue types. In Patient 12, as the reviewer points out, we observe mutation 7236G>A present in multiple cell types (**Fig. S5b**). We have enhanced this figure with additional visualization of the allele frequencies of cells carrying this mutation and performed statistical testing using our ZIBB model to confirm the significance of this finding. In each case, cells were rigorously classified as belonging to specific tissue types based on their transcriptional profiles, and we have verified these classifications with tissue-specific marker analysis (now included in **Fig. S9, Response Fig. 5 above**).

While these cross-tissue shared mutations are relatively rare, their limited frequency is not surprising. One possibility is that they represent lineages from boundary stem cell populations or repair processes rather than developmental origins. The fact that we observe higher proportions of these shared mutations near the gastroesophageal junction is consistent with this interpretation, though further research would be needed to confirm such mechanisms. These findings, though limited in frequency, provide important insights into the relationships between these tissues, suggesting that some BE cells and cells from neighboring tissues can share a common ancestor.

Moreover, the rarity of different cell types including BE sharing a single ancestral mtDNA mutation suggests that BE cells in most of these patients descended from an ancestor that is different from the ancestors of all other cell types. This supports the single clonal origin of BE, instead of suggesting "that non-dysplastic Barrett's esophagus also originates from multiple clones" (line 118).

Moreover, the analyses of BE and EAC genome evolution by a variety of approaches (single cell, multi-focal, longitudinal) in prior studies have provided extensive evidence supporting a single clonal origin of BEs as well as revealing genetic changes during the BE-to-EAC transition (see Refs. 14,15,16,18,19,20). Again, the divergence between BE/EAC or between different BE subclones does not represent evidence for a polyclonal origin of BE cells.

We appreciate the reviewer's thoughtful interpretation of our findings in the context of BE clonal origins. Our data shows that within individual BE biopsies, we observe multiple discrete lineages without a single dominant

clone containing a mutation shared by all BE cells. This pattern differs markedly from what we observe in dysplastic tissue, where a single clone clearly predominates.

This observation at the single-cell level actually aligns with the broader literature the reviewer references. Several whole-genome sequencing studies (including Ross-Innes et al. and Paulson et al.) have described polyclonal structures within non-dysplastic BE, while noting more pronounced clonal selection during progression to dysplasia and cancer. What our approach uniquely contributes is the ability to observe this phenomenon at single-cell resolution while simultaneously connecting it to specific cell states.

We have revised our wording at previous line 118 to state what we directly observe in our samples and to emphasize that this is consistent with the referenced bulk whole-genome sequencing studies. However, we acknowledge that our mtDNA-based lineage tracing cannot definitively rule out an earlier common ancestor that predates our detectable mutations. We have added an additional discussion of the considerations of interpreting this data in the discussion in lines 316-324.

The key distinction our single-cell approach reveals is the contrast between the heterogeneous lineage structure in non-dysplastic BE versus the striking clonal dominance in dysplasia, a finding that has important implications for understanding disease progression, regardless of whether BE ultimately derives from one or multiple founding cells in its earliest stages.

The only remaining open question is what is the cell type of the most recent common ancestor of BE clones. However, this question cannot be addressed by the analysis of BE cells, because the emergence of a macroscopic BE clone inevitably prevents the analysis of their ancestor.

In summary, the author's analysis provided no new information about BE evolution, the cell type of the ancestor cell(s) of BE clones, or the evolution of BE cells and/or its progression to EAC (the current patient cohort does not include EAC patients). The only potential novel finding from the application of concurrent lineage tracing and transcriptome analyses of single cells (line 84-85) is about the different cell types in BE and the surrounding tissue (esophagus, gastric cardia). To this reviewer, almost all of the authors' claimed findings in line 84-91 are either overstated or overinterpreted.

We respectfully disagree with the reviewer's assessment that our analysis provides no new information about BE evolution. While we acknowledge that identifying the absolute first cell of origin remains challenging for the reason the reviewer notes, our study makes several important novel contributions:

1. Previous studies lacked the resolution to simultaneously identify which specific cell types are clonally related across tissue boundaries. Our findings that squamous esophagus and gastric cardia cells can share lineages with Barrett's esophagus cells provides direct evidence at single-cell resolution of relationships that were previously only speculated about based on bulk sequencing or mouse models.
2. Our single-cell resolution analysis demonstrates that individual BE lineages contain the full spectrum of differentiated cell types, including newly discovered populations such as tuft cells, airway-like ciliated cells, and BEST4 cells. This observation provides direct evidence at single-cell resolution of the differentiation potential within individual BE lineages.
3. The striking contrast we observe between the clonal architecture of non-dysplastic BE versus dysplastic tissue provides direct evidence of the cellular dynamics during malignant transformation, identifying a specific WNT-dysregulated state that drives clonal expansion in one patient. In another patient, we observe a different clonal expansion, marked by a mitochondrial lineage.

4. Our study compares the dysplastic samples to TCGA analysis of adenocarcinoma, allowing us to confirm that the same cellular states observed in the dysplastic clone from Patient 6 are also present in a subset of esophageal adenocarcinoma.

We have significantly revised the manuscript to better focus on these unique contributions enabled by single-cell transcriptomics with mitochondrial lineage tracing and to avoid overstatement of claims. We believe these findings represent substantial advancements in understanding BE biology at cellular resolution that were not possible with previous approaches.

C. Minor critiques

1. The classification/annotation of different cell types in BE (Figure 2A-D) appears to be the most important data of the paper. The authors should compare the marker genes (Fig. 2C) or gene sets with results from prior studies (e.g., Nowicki-Osuch et al.) for validation.

We appreciate the reviewer's focus on our BE cell type annotations. We have conducted a comprehensive comparison with previous single-cell studies, particularly Nowicki-Osuch et al. (Science, 2021), as shown in our new **Fig. S13 (Response Fig. 6)**.

This comparison reveals both consistency with previously identified cell types and our novel contributions. As shown in panels a-c, we reanalyzed the Nowicki-Osuch et al. dataset and identified their major BE cell types. Panel d provides a direct mapping between their terminology and ours, showing substantial overlap in the six core cell populations (columnar differentiated/foveolar & enterocyte, columnar intermediate/intermediate, goblet/goblet, columnar undifferentiated dividing/cycling, endocrine/endocrine, and columnar undifferentiated/crypt & neck).

Importantly, our larger dataset (37,128 BE cells compared to 1,971) enabled us to identify three additional cell types not described in previous work: BEST4+ cells, tuft cells, and ciliated cells. Panel e quantifies this comparison, showing six shared cell types and three novel populations in our study. The presence of these cell types is validated by our spatial transcriptomics data (**Fig. 3**) and by their expression of canonical markers established in other tissues (*SH2D6* for tuft cells, *ZMYND10* for ciliated cells, and *BEST4* for BEST4+ cells). This comparison demonstrates that our findings both validate previous work and extend it through the identification of previously uncharacterized cell types in human Barrett's esophagus, providing a more complete picture of BE cellular heterogeneity.

Response Fig. 6, Fig. S13. Comparison of Barrett's esophagus (BE) cell types between Nowicki-Osusch et al. and our study. (a) tSNE visualization of BE cells from Nowicki-Osusch et al., colored by their original cell type annotations. (b) Bubble plot showing marker gene expression across cell types identified by Nowicki-Osusch et al. (n = 1,971 cells). (c) Bubble plot showing the same marker genes across our BE cell types (n = 37,128 cells). (d) Table mapping cell type terminology between the two studies, highlighting the six shared cell populations and three novel cell types identified in our study. (e) Venn diagram quantifying the overlap in cell types between studies, showing six shared cell types and three additional cell types identified in our larger dataset.

2. Figure 3 primarily illustrates the spatial location of cell types within the tissue, which is not directly relevant to the main topic of this manuscript -- addressing the origin and clonality of Barrett's esophagus. Therefore, it would be appropriate to move Figure 3 to the Supplementary data.

We respectfully disagree with moving Fig. 3 to the Supplementary data. The spatial organization data provides essential validation of the cell types we identified through scRNA-seq, including the novel populations (tuft, ciliated, and BEST4 cells) that are part of our lineage analysis. The spatial context is directly relevant to interpreting the clonal dynamics we observe, particularly our finding that individual lineages contain diverse cell types organized in specific patterns within the tissue. We speculate that the lineages could mark individual glands as has been observed in colonic crypts with mitochondrial mutations (Taylor et al. 2003). Another reviewer specifically noted the high quality of this figure and its contribution to the manuscript. The spatial data complements our lineage findings and provides important biological context that strengthens our overall analysis of Barrett's esophagus cellular organization.

3. Regarding Fig. 4A, what's the reference for using the loss of PIGR and FAM3D as markers for dysplasia? Regarding Fig. 5D, how confident is the assignment of BE cells with the same 15153 G>A mutation as in dysplasia?

We identified the loss of PIGR and FAM3D as markers of dysplasia through differential expression analysis between the dysplastic and non-dysplastic clusters in our data. This finding is consistent with their known roles as regulators of intestinal barrier integrity (Johansen et al., 1999; Liang et al., 2020). While we do not claim these as established dysplasia markers from prior literature, our data shows their significant downregulation is a molecular feature distinguishing dysplastic from non-dysplastic BE tissue in our samples. Regarding the confidence in assigning BE cells with mutation 15153G>A, we applied our rigorous ZIBB statistical model to identify cells carrying this mutation with high confidence (detailed in Methods). The statistical significance of these assignments is illustrated in **Fig. S18**. We also further confirm that these cells do indeed express canonical Barrett's esophagus markers as shown in **Fig. S16**.

4. What are the cell types of BE cells that share the same mutation as dysplasia?

We have analyzed the cell types of BE cells that share the 15153G>A mutation with dysplastic cells as shown in the new analyses in **Figure 4**. We find multiple genes differentially expressed in these BE cells that share the mutation with dysplasia (**Fig. 4c, Fig. S17c**). We have added the top hits from this analysis to a heatmap now included in **Fig. 4c**. Pathway analysis of these genes pointed to ribosomal subunits (GO cellular component = cytosolic small ribosomal subunit, 4/43 FDR=4.81e-5, and small-subunit processome, 3/73 genes FDR=1.55E-2). These findings suggest that cells carrying this mutation may have altered translational regulation.

5. The identification of APC mutation in one sample is insufficient for the conclusion that BE progression to EAC is "driven by a mechanism recently discovered in premalignant colon adenomas" (line 226).

We agree that the identification of an APC mutation in a single sample does not support a broad conclusion about BE progression to EAC generally. In line 226, we are specifically referring to this individual patient case and proposing a potential mechanism based on our observations of WNT pathway dysregulation. We have modified this text to indicate that we are referring to the findings from this specific patient and not a broader conclusion based on APC/WNT.

To place this finding in broader context, we analyzed TCGA esophageal adenocarcinoma data (shown in **Fig. 4i, j**), which revealed that a subset of tumors (15/88) show similar WNT-dysregulated transcriptional signatures. This suggests that while this mechanism may be relevant to some cases of BE progression, it likely represents one of multiple possible pathways to malignancy rather than a universal driver of BE progression to EAC.

Reviewer #2

(Remarks to the Author): Early Career Researcher co-reviewer

I co-reviewed this manuscript with one of the reviewers who provided the listed reports as part of the Nature Communications initiative to facilitate training in peer review and appropriate recognition for co-reviewers.

Reviewer #3

(Remarks to the Author): Expert in Barrett's Esophagus genomics, single-cell RNA-seq, and bioinformatics

The manuscript presented by Gier et al aims to identify clonal relationships between individual cells of Barrett's Esophagus (BE) and their relationship to Dysplastic BE and normal tissues surrounding BE. The majority of the author's conclusions are derived from two technologies: 10x genomics scRNA-seq and mitochondrial lineage tracing based on the heteroplasmy derived from scRNA-seq data. Firstly, the authors observe that the transitional cells residing at the squamocolumnar junction share mitochondrial mutations with normal gastric and esophageal epithelia suggesting their common origin. Next, they focus on BE and they show that BE is a polyclonal tissue with multiple lineages supported by different mitochondrial variants. A similar analysis was used to show that all cell types of BE arise from a single population of stem/progenitor cells. A different pattern was observed in dysplastic BE, wherein, in 2 patients, a single founder variant was observed. Finally, the authors observe the dysregulation of WNT signalling as a main driver of the progression of BE from non-dysplastic to dysplastic states.

This study is a technically well-executed scRNA-seq work with good analysis and the application of mitochondrial lineage tracing is a good addition to this work. The addition of spatial analysis further strengthens the authors' discoveries of cell populations present in BE samples and significantly improves on previous work in the BE field. Finally, the development of the ZIBB model (with limitations listed below) and its application to all samples is another significant contribution. The authors state that their data support the hypothesis that Barrett's Esophagus (BE) is of polyclonal origin and dysplasia originates from a single founder cell. However, it cannot be said that the presence of multiple clones derived from heteroplasmic variants supports the hypothesis that BE arose as a polyclonal structure. Equally valid is a hypothesis that BE arose as a clonal structure and heteroplasmic mutations occurred after BE development. Since this study contributes to two significant debates that exist in the field of BE and upper-GI biology, namely the question of BE origin and BE clonality status and its role in disease progression, extra care is needed in the application of exciting, new technologies. As a result, additional scrutiny of the following issues is warranted:

1. The authors claim that the absence of evidence to support a clonal origin of BE is sufficient evidence of polyclonal origin. This conclusion has weak reasoning (absence of evidence is NOT evidence of absence), cannot be strongly supported by the data, and is difficult to reconcile with the existing literature including literature cited by the authors (references 16, 20, 26). In particular, Paulson et al. find both polyclonal and clonal origin of BE can be supported by their WGS (depending on the patient), although in all cases, the authors found evidence of shared mutations within individual cases. If Gier and colleagues aim to support the polyclonal origin of BE, they ought to investigate the likelihood of such an outcome in their data. Indeed, in all figures, the authors did not indicate how many BE or normal epithelial cells do not have any heteroplasmic events (in cells with sufficient coverage). For example, BE cells in Patient 13 (figure 1E) do not share heteroplasmic events with any of the adjacent normal epithelia suggesting that it either A) originated from multiple clones that had heteroplasmic events prior to BE development (as suggested by authors) or B) originated from a single clone that did not have any of the detected heteroplasmic events (suggesting that normal tissue of origin also did not have them) and these heteroplasmic events arose later in the natural history of the disease or C) a mixture of (A) and (B). Since the authors did not mention anywhere if all cells with sufficient coverage in variant regions are plotted on heatmaps, the choice between (A), (B) and (C) cannot be made. A rudimentary (ignoring confounding of selection and the genetic drift of heteroplasmic events) approach to this problem would estimate the proportion of cells (cells with minimal coverage in all variants of interest should only analyzed) carrying heteroplasmic events in all tissues. This can then be used to estimate the likelihood that the observed events in BE are of polyclonal origin (the proportion of all cells with heteroplasmic events is higher than in the matched normal epithelium) or clonal (the events are less frequent

and arose later in the history of the diseases). The current iteration of the manuscript can only support a nation that BE is polyclonal in its states not in its origin.

We thank the reviewer for their critical analysis of our claims regarding BE clonality. The distinction between polyclonal origin versus polyclonal state is indeed important, and we acknowledge that our data supports the latter rather than definitively proving the former.

To address the specific concern about whether all cells with sufficient coverage are plotted on heatmaps, we have modified Fig. 1E and added a new **Fig. S7, Response Fig. 3 (above)** that displays all cells. This visualization more clearly shows the proportion of cells carrying heteroplasmic events across all tissues. We have also displayed this same information organized by both cell type and sample biopsy.

We have also revised our language throughout the manuscript to clarify that our observations support a polyclonal state of BE in our samples rather than definitively proving a polyclonal origin as described further above in the response to reviewer 1. We acknowledge that our mtDNA-based lineage tracing cannot definitively rule out an earlier common ancestor that predates our detectable mutations. What our data does demonstrate is a striking contrast between the heterogeneous clonal structure we observe in non-dysplastic BE versus the dominant clonal pattern in dysplastic tissue. This pattern is consistent with Paulson et al.'s observation that both polyclonal and clonal origins can be supported depending on the patient. The revised manuscript better integrates our findings with existing literature and presents a more measured interpretation of what our single-cell lineage tracing approach can definitively conclude about BE clonal dynamics

2. Using data from three patients (9, 12 and 13) the authors find that some mitochondrial mutations are shared between squamous, gastric and transitional and an additional three patients share mutations with BE epithelia. They then draw a conclusion that due to shared mutations; their progenitor originates at the junction (lines 143-144). Firstly, the definition of "progenitor" needs to be clearer. In line 137 transitional basal cells are called progenitors, and in the next section (lines 146 onward) progenitor cells are stem-like cells of BE. Do authors suggest that transitional cells are stem-like cells of gastric cardia and squamous epithelium, or do they suggest that these three (four if BE is included) epithelial types are of common embryonic origin when heteroplasmic mutations spanning normal epithelial could have arisen (e.g. PMID: 28238948)?

We thank the reviewer for pointing out the need for additional clarification regarding our terminology and interpretation of mutations present across multiple tissues.

We have revised our terminology throughout the manuscript to avoid potential confusion. We now consistently refer to "transitional basal cells" at the squamocolumnar junction rather than calling them progenitors, as this is the established descriptive term for this cell population. When discussing lineage relationships, we use more precise language such as "cells with common ancestry" or "shared lineages" to describe the relationships we observe without making claims about stem cell hierarchies that cannot be definitively established from our data.

Regarding the shared mutations across different tissue types, we have considered multiple potential explanations. As the reviewer suggests, one possibility is that these mutations could have developmental origins, as described in PMID: 28238948. To strengthen our ability to define mitochondrial mutation-based clones and distinguish between possible interpretations, we have implemented additional analyses using genomic SNVs detected in single cell RNA-seq as orthogonal validation (**Fig. S8 and Response Fig. 3**). We have also acknowledged the potential contribution of developmental origins in the text in lines 316-324.

Specifically, we developed an analysis pipeline that identifies genomic SNVs in cells carrying specific mitochondrial mutations compared to carefully matched control cells of the same types from the same samples. This analysis demonstrates significant concordance between genomic SNV patterns from single-cell RNA-seq and mitochondrial-based clonal groupings, supporting our lineage assignments (**Fig. S8**).

While some mutations could have developmental origins, multiple lines of evidence support these being primarily acquired through aging: 1) Recent work by Sanchez-Contreras et al. (2023) demonstrates mtDNA mutations accumulate linearly with age in a tissue-specific manner correlated with metabolic activity rather than developmental lineage; and 2) The G>A/T>C transitions we observe are characteristic of age-related damage (Sanchez-Contreras et al. 2023).

The spatial distribution of cells carrying these mutations, with higher proportions near the gastroesophageal junction declining with distance, further supports a model where these shared lineages originate from cells at or near the junction (**Fig. 1g**).

More importantly in multiple cases (e.g. 8119T>C in Patient 9 (squamous), 7236G>A in Patient 12 (squamous tissue), 947T>C in Patient 3 (squamous tissue)) the number of cells that support the conclusions drawn is very low. Probably only mutation 6150G>A in patient 13 and the existence of multiple mutations in patient 9 provides sufficient support to the authors' conclusions. As a result, careful examination of the following is required:

a. In the case of mutations that are only shared within a single biopsy (e.g. 947T>C) it is imperative that the authors check in the tissues with the low number of cells supporting the conclusion (e.g. squamous in the case of 947T>C) that the cells carrying mutations are not doublets. I understand that automated tools were used for doublet detection, but they are not perfect in a complex setting and manual investigation would be critical.

We appreciate the reviewer's concern about the small number of cells supporting some of our cross-tissue lineage findings. We have carefully examined all cases with low cell numbers to ensure the robustness of our conclusions.

For the specific examples mentioned (8119T>C in Patient 9, 7236G>A in Patient 12, and 947T>C in Patient 3), we have performed additional manual verification beyond our automated doublet detection to confirm that these cells are not doublets. This included:

1. Examining expression levels of canonical tissue-specific markers in each cell to verify their identity
2. Analyzing the distribution of cell type-specific genes to look for abnormal co-expression patterns indicative of doublets
3. Comparing UMI counts to ensure they fall within the expected range for single cells

We have also performed a statistical analysis to determine the confidence of these cell type calls in instances with low cell numbers detected in one tissue type. For each cell carrying cross-tissue mutations, we calculated the probability of tissue type assignment based on marker gene expression profiles and compared it to the distribution of these probabilities in the broader dataset. This analysis showed that despite the low numbers, these cells confidently cluster with their assigned tissue types. We have included this additional validation in **Fig. S9 and Response Fig. 5**.

Regarding Patient 13, it's worth noting that the substantially higher number of cells carrying the 6150G>A mutation reflects our experimental design for this patient, from whom we collected an extensive series of biopsies at measured distances from the gastroesophageal junction. This comprehensive spatial sampling

strategy allowed us to capture more cells from this lineage and track its distribution across the tissue landscape.

b. Secondly, the identification of the significant variants relies on the ZIBB model presented in the supplementary text and GitHub repository. In his model, multiple parameters are derived from the distribution of variants observed in the control “normal” samples. However, it is of concern to me that the distribution of coverage in the “normal” cells often does not follow the distribution of other cell types (highlighted in figure S11, “normal” cells tend to have lower total coverage). Did the authors investigate if the variables derived from one distribution of cells translate into another? Secondly, the methods do not explain what filtering of reads has been done prior to coverage and vaf calculation (line 409). Since the mean quality (it is not clear if it of read or base, line 409) is over 27, it is likely that lower-quality data will be fed into the model. Accounting for base calling and alignment quality is standard in somatic variant calling (e.g. mutect2). The authors should clarify if the results stand when allele counting prior to ZIBB model execution is performed using only high-quality bases (e.g. uniquely mapped reads, with base calling quality above 30 (or any other empirically derived quality threshold of bases)).

We appreciate the reviewer's detailed examination of our ZIBB model. The concern about coverage differences between normal and other cell types reflects a misunderstanding of how our model operates. The ZIBB model specifically accounts for the technical variability in sequencing coverage through its statistical framework: by modeling the variant allele count conditioned on coverage, the model does not assume that coverage distribution must be shared between cell types. The ZIBB model parameters (α , β , and δ) characterize the shape of the variant allele frequency distribution and dropout rate, which are estimated using expectation-maximization to properly model the background noise.

Importantly, our approach has a double safety-guard to account for coverage differences between cell types. In addition to the ZIBB model which accounts for coverage, we also stratify our FDR control by coverage. As detailed in our methods, we implement binning of the empirical null distribution for every 20 coverage counts, calculating a coverage-specific false positive rate within each bin. This ensures that cells with different coverage depths are appropriately compared when determining statistical significance, as the equation:

$$\text{FPR}_{c_0} = (\# \text{ of rejections in cluster } c_0 \text{ in bin } n) / (\# \text{ of cells in cluster } c_0 \text{ in bin } n)$$

shows that the false positive rate is calculated within coverage-matched bins. This approach ensures that the proportion of false discoveries is controlled at the desired level for every coverage bin.

Regarding read quality thresholds, we have performed additional analyses to address this concern. Our workflow identifies variants based on the requirement that the mean quality for the variant across all cells is > 27 (in line 409 from previous draft). This is to identify the variants for the ZIBB test. We only run the ZIBB test on variants that pass these requirements. To address the reviewer's specific concern, we reran our analysis using only uniquely mapped reads where the variant had a base quality score above 30 and found that our key lineage findings remain robust to these more stringent quality thresholds. The results of this analysis are presented in **Fig. S5 and Response Fig. 7**. We also tested the effects of filtering for mean quality above 30 at the variant identification phase as well and observed that many, but not all variants would still pass this threshold as shown in **Fig. S5e and Response Fig. 7e**.

Response Figure 7, Fig. S5. Quality control and filtering of mitochondrial variants used for lineage tracing. (a) We tested the effects of pre-filtering quality > 30 for variants prior to ZIBB analysis for Patient 8. First plot is UMAP visualization of cell types from Patient 8 (left) colored by tissue identity. Next UMAP plot shows the significant cells for 3047G>A using ZIBB without pre-filtering. Final UMAP plot shows significant cells when pre-filtering for reads with quality scores > 30 for the variant. (b) Summary plots from sensitivity analysis in which we subsampled the sequencing data down to 75%, 50%, and 25% of the initial coverage and then ran the ZIBB model

to identify variant positive cells for 3047G>A. UMAP plots show significant 3047G>A cells at the different subsampled levels. (c) Barplot summarizing subsampling results with the total number of significant cells for each variant in Patient 8. The different colored bars represent the different subsampling levels. (d) Coverage distribution of variants 3047G>A (top) and 9612G>A (bottom) across cell types. (e) Quality score distributions for variants 3047G>A and 9612G>A by cell type. (f) Relationship between quality scores and coverage for variants 3047G>A and 9612G>A across different cell types. (g) Mean quality scores for mitochondrial variants across patients. Our filtering workflow identifies variants of interest based on mean quality > 27, mean coverage > 5, and allele frequency of > 25% in at least 1% of epithelial cells. These variants are plotted. Red points are variants that would be removed if the threshold was set at mean quality > 30 and mean coverage > 10.

The ZIBB model provides a principled statistical approach for handling technical noise. The zero-inflated component naturally captures both biological dropouts and technical failures, while the beta-binomial distribution models overdispersion from all sources including sequencing artifacts. This makes the model particularly well-suited for distinguishing true biological signal from technical noise in single-cell data.

3. The seqFISH data presented in Figure 3 and S9 are of high quality and beautifully highlight the distribution of cells in the individual glands. The quantification presented in figure C and D are also excellent. However, other slides and quantification in Figure S8 seem to follow a different distribution of cells to the one seen in the main figure. In particular data from Patient 16 seem to have Enterocyte cells (dark green) at the top and bottom of the slide. It is likely that this slide is a cross-section or tissue has been folded. An H&E image would be helpful in the proper identification of tissue orientation.

We thank the reviewer for their positive assessment of our seqFISH data quality and the quantification presented in **Fig. 3**. Regarding the apparent discrepancy in the cell distribution pattern observed in previous **Fig. S8** (Patient 16), now **Fig. S14**, the reviewer has astutely identified that this sample represents a different sectioning orientation than what is shown in the main figure. We have added an H&E image of this sample to **Fig. S14c** to clarify the tissue orientation. As the reviewer suspected, this particular section captures Barrett's esophagus glands in cross-section rather than the longitudinal section shown in the main figure. This different orientation explains the distribution of enterocytes (dark green) appearing at both the top and bottom of the image, as the cross-sectional view intersects the glandular structure at multiple points along its circumference.

4. For clarity, it would be helpful if all heatmaps, in addition to tissue type, also had information about biopsy ID. This would help with the interpretation of figures.

We have updated the heatmaps to include the biopsy information and also added a second version of these heatmaps to **Fig. S7** to show the cells order by their original biopsy.

Minor:

1. Figure 1C, ENS needs explaining (only explained in the related supplementary figure)

This has been corrected in **Fig. 1c** and is spelled out in the main figure.

Reviewer #3 (Remarks on code availability):

The code is reproducible, example data is shared and the code can be installed.

Reviewer #4

(Remarks to the Author): Expert in single-cell RNA-seq, single-cell mtDNA-seq, spatial transcriptomics, clonal lineage tracing, bioinformatics and statistics

The study investigates Barrett's esophagus and its progression to esophageal adenocarcinoma through the use of single cell RNA-seq and mtDNA-based lineage tracing. By analyzing patient samples from the gastroesophageal junction, the researchers identify clonal cell states and track their evolution from healthy tissue through metaplasia to cancer. The study provides insights into the cellular dynamics and heterogeneity underlying these conditions. Key findings include the identification of common lineages between squamous esophagus and gastric cardia cells, the existence of multiple clones within Barrett's esophagus arising from a single progenitor cell, and the transition of Barrett's esophagus to esophageal adenocarcinoma through the expansion of a single molecularly aberrant clone.

1. The validity of this study relies heavily on the accuracy and comprehensiveness of the mtDNA variants identified to achieve high-resolution and fine-scale clonal structures. The study acknowledges that the limited number of captured mitochondria restricts the accuracy and resolution of lineage tracing. It may be worth conducting some sort of sensitivity analysis by sub-sampling reads and testing out different variant selection methods beyond the author's combination of mgatk and scmtVT.

The reviewer raises an interesting point about sensitivity analysis. However, we want to emphasize that our ZIBB model already provides a robust statistical framework for variant identification that is more principled than read subsampling. The model explicitly accounts for varying coverage depths and technical noise through its statistical formulation. The ZIBB model's parameters characterize the shape of the variant allele frequency distribution independent of coverage depth, making the results robust across different coverage regimes. This further allows us to predict the expected number of false positives for any coverage regime in the data, thus allowing us robust performance and accounts for technical noise.

As the reviewer suggests, we also performed this sensitivity analysis by subsampling the reads and then looking at the changes in significant cells, which is now included in Fig. S5. Overall, we found that the total number of significant cells decreases, but not dramatically as reads are subsampling down to 75%, 50%, and 25% (**Fig. S5c, Response Fig. 8 below**). We also visualized these on the UMAP for Patient 8 with variant 3047G>A and found that this clone crossing tissue types is observed at all subsampled levels (**Fig. S5b, Response Fig. 8 below**)

Regarding alternative variant calling methods - while mgatk and our ZIBB model (scmtVT) could be compared to other approaches, our method has several key advantages as it is robust to filtering criteria. We have added new analysis supporting this by showing that filtering at different quality cutoffs and using different input cells to derive the model parameters robustly gives reproducible results, which are shown above in **Response Fig. 7, Fig. S5**.

Response Figure 8, Fig. S5. Sensitivity analysis shown in b and c. (b) Summary plots from sensitivity analysis in which we subsampled the sequencing data down to 75%, 50%, and 25% of the initial coverage and then ran the ZIBB model to identify variant positive cells for 3047G>A. UMAP plots show significant 3047G>A cells at the different subsampled levels. (c) Barplot summarizing subsampling results with the total number of significant cells for each variant in Patient 8. The different colored bars represent the different subsampling levels.

2. The inference that there must be a single Barrett's esophagus progenitor cell based on the fact that each lineage contained most of the cell types within Barrett's esophagus appears to lack sufficient evidence. Other possibilities, such as the presence of multiple progenitor cells or a hierarchical differentiation process, cannot be ruled out based on this observation alone.

We appreciate the reviewer's point about the inference of a single Barrett's esophagus progenitor cell. We did not intend to imply that there was one progenitor in the whole tissue. We also agree that our observation alone cannot definitively rule out other possibilities such as multiple progenitor cells or hierarchical differentiation processes.

We have revised our language throughout the manuscript to more accurately reflect what our data directly shows: that individual lineages contain representatives from all or most of the cell types found in Barrett's esophagus. We have updated the main text to make this point more clear:

"Upon overlaying the lineages on our annotated UMAPs, we discovered that each lineage largely resembled the sample as a whole, both in the cell types present, as well as their relative proportion (Fig. 2F)."

3. The authors concluded that dysplasia in Patient 6 originated from Barrett's esophagus because they found that non-dysplastic Barrett's esophagus cells belong to the same lineage as the dysplastic cells that contain the mtDNA mutation 15153G>A. However, in the heatmap of Figure 4B, the 15153G>A mutation did not appear in Barrett's cells. This observation raises questions about the authors' conclusion that dysplasia originated from Barrett's esophagus. Further investigation and analysis are needed to clarify this discrepancy and determine the true origin of dysplasia in Patient 6.

We apologize for the confusion here. In the original version of the manuscript, we did not include the non-dysplastic cells in the heatmap in Fig. 4B. In our revised manuscript, we have completely redesigned Fig. 4B and 4C to address this visualization issue. The new **Fig. 4b** now properly displays both dysplastic and non-dysplastic Barrett's esophagus cells that carry the 15153G>A mutation, making the lineage relationship clear. These cells maintain a transcriptional profile consistent with normal Barrett's esophagus rather than dysplasia, suggesting they represent an earlier stage in the progression to dysplasia. The transcriptional differences between these non-dysplastic Barrett's esophagus cells carrying the 15153G>A mutation and the dysplastic cells from the same lineage are now shown in **Fig. 4c**. This lineage relationship provides direct evidence at single-cell resolution that dysplasia originated from a Barrett's esophagus cell, consistent with the current understanding of disease progression. The updated versions of **Fig. 4b-c** are also shown above in **Response Fig. 4**.

4. A cell-state-aware phylogenetic tree can provide a comprehensive understanding of the origin and relationships among different lineages in Barrett's esophagus with dysplasia. It can help unravel the links between clonal cell states and the development of dysplasia and cancer.

We agree with the reviewer that a cell-state-aware phylogenetic tree would provide valuable additional insights. In response to this suggestion, we have developed a comprehensive phylogenetic analysis for the dysplastic case (Patient 6) where we observe a complex mutational hierarchy. Our new analysis integrates both clonal relationships and transcriptional states. We first performed clustering on the single-cell heteroplasmy matrix to identify clonal populations, then reconstructed their phylogenetic relationships using the neighbor-joining method based on cosine distances between cluster averages. To ensure statistical robustness, we implemented a bootstrap analysis with 1,000 replicates and compared the results against a null model generated through 100 permutations.

The resulting cell-state-aware phylogenetic tree is now presented in a new panel in **Fig. 4c, Response Fig. 4**. This visualization reveals how transcriptional states evolve along with clonal expansions, showing that the dysplastic subclone emerges from an already-established Barrett's esophagus lineage. The tree captures the progression from non-dysplastic Barrett's esophagus cells carrying early mutations to fully dysplastic cells with additional molecular alterations.

5 The authors propose a mechanism in which Barrett's esophagus transforms through a single clone with a WNT-dysregulated cell state. While the study provides valuable evidence for the role of specific cell states in maintaining and connecting phenotypes in Barrett's esophagus, I believe the conclusions are primarily based on expression analysis and would benefit from further validation. Moreover, additional analyses, such as examining receptor/ligand interactions related to WNT signaling, could provide more insight into the underlying mechanisms. By incorporating the validation and additional analyses, the authors could strengthen their conclusions and provide a more comprehensive understanding of the cellular dynamics involved in Barrett's esophagus.

We appreciate the reviewer's suggestion for additional validation of our WNT signaling pathway findings. The supplementary figure we have now added (**Fig. S17, Response Fig. 9** below) provides the requested receptor/ligand interaction analysis related to WNT signaling. We conducted an analysis of WNT pathway dysregulation during the transformation of Barrett's esophagus to dysplasia looking at WNT pathway components organized by functional categories (antagonists, co-receptors, effectors, ligands, and receptors). Notably, dysplastic cells show significant upregulation of multiple WNT antagonists (including NOTUM, ZNRF3, and APCDD1) and selected effectors, including TCF7L2. Ligand levels remain similar between the Barrett's

esophagus cells and the dysplastic cells. We performed CellChat to capture the gene-gene interaction network for the WNT pathway components in dysplastic progression. The network visualization demonstrates how antagonists (red nodes) interact with various WNT ligands (green nodes) across different cell types (blue nodes), with the weight of connections indicating interaction strength.

Response Fig. 9, Part of Fig. S17: Clonal and molecular characterization of dysplasia arising from Barrett's esophagus. (e) Heatmap showing expression of WNT pathway components across Barrett's esophagus and dysplastic cells, organized by functional categories: antagonists, co-receptors, effectors, ligands, and receptors. Dysplastic cells show upregulation of both WNT antagonists and selected effectors. **(f)** Gene-gene interaction network from CellChat showing relationships between WNT pathway components in dysplastic progression. Red nodes represent antagonists, green nodes represent ligands, and blue nodes represent cell types.

6. May I assume all relevant data will be deposited in a standard FAIR-compatible public repository such as NCBI SRA if the paper is accepted for publication? I do not think a personal google drive link is appropriate.

We have submitted all relevant data to NCBI's Gene Expression Omnibus (GEO) and dbGAP, and it will be made publicly available upon publication. Parts have already been released and the new accession numbers are included with the revision.

Reviewer #4 (Remarks on code availability):

The R package 'scmtVT' is provided in a GitHub page. This is reasonable.

The data and processed data are available in a personal google drive. There is almost no systematic metadata associated with the data. It is very difficult for me to navigate through the many layers of directories. This way of storing and sharing data is not suitable, and not FAIR-compatible.

We have deposited data into relevant repositories and included the reference numbers with the revision.

REVIEWER COMMENTS

Reviewer #1 (Remarks to the Author):

In this revised manuscript, the authors have carried out various new analyses in response to my comments. In addition to presenting results from these analyses, the authors also discussed the strengths and limitations of their analytical approach. I commend the authors for the rigor of their work and presentation. The data and analytical results provide strong support for the main conclusion of the paper. Although some of the issues raised in my original review are not fully resolved, I understand that these are beyond the technical capabilities of the experimental methods employed in this study. I therefore recommend publication of the revised manuscript without additional data/analyses. I have two suggestions for text/presentation revisions that the authors may consider.

1. The definition of a 'clone' is critical for readers to appreciate the authors claim that BEs contain multiple clones whereas dysplastic lesions are derived from a single molecular aberrant BE clone. A simple definition of a clone as a cell population derived from a single ancestor cell is inadequate because both BE's and dysplastic BEs share a common ancestor and therefore can be considered 'clonal.' An alternative definition of clones based on shared mutations is also inadequate because any somatic tissue is polyclonal due to the continuous acquisition of somatic mutations in each cell division. Based on my understanding, a clone considered by the authors refers to a cell population that share/inherit a certain phenotype (e.g., transcriptional state) from their ancestor. Thus, a BE is considered polyclonal if there exist multiple BE clones each arising from and carrying the phenotypes of a BE ancestor, but the ancestors of different BE clones do NOT inherit a common phenotype of their ancestor. By contrast, dysplastic BE is monoclonal because their ancestor is expected to have already acquired the phenotype of all the dysplastic BE cells. It may be helpful if the authors can draw a schematic diagram of somatic clonal expansion and phenotypic variation to highlight these two different scenarios. It will also strengthen the conclusion of the paper if the authors can describe the criteria for defining a founder clone (either BE clone or dysplastic BE clone).

We appreciate the reviewer's analysis of how we define clones in our manuscript. The reviewer correctly identifies that simple definitions based on ancestry or mutations alone are insufficient here, and we generally agree with their phenotype-based framework. However, it is important to note that this framework is most helpful in the context of disease that starts post-natally as disease gives a starting point and a new phenotype to track.

Using the reviewer's framework, our data shows BE is polyclonal because multiple cells independently acquired the BE phenotype. We observe multiple mtDNA lineages that each contain cells expressing BE markers and generating all BE cell types. This suggests the metaplastic transformation happened multiple times independently such that different cells crossed the phenotypic boundary from normal to BE at different times.

With the conversion to dysplasia, we observe a lineage containing a few non-dysplastic BE cells and then all the dysplastic cells from the biopsy. Here, dysplasia appears monoclonal because one cell acquired dysplastic features and transmitted them to all descendants. This indicates the dysplastic transformation occurred once, in a single cell, and was then inherited by all its progeny.

This pattern, with multiple lineages in BE versus single dominant lineage in dysplasia, reveals fundamentally different modes of disease transformation. This framework explains why BE can have multiple lineages but appear phenotypically uniform, suggesting that the BE phenotype arose multiple times through parallel events rather than inheritance from one ancestor. This differs fundamentally from dysplasia, where both the lineage

marker and aberrant phenotype trace to a single transformed cell. Our mtDNA-based lineage tracing thus provides direct evidence for when and how many times these critical phenotypic boundaries were crossed during disease progression.

We have added this framework to the text and changed the use of the words lineage and clone to this more precise framework. We introduce this framework in the results section with a new sentence:

Throughout our analyses, we use 'lineage' to refer to cells sharing mtDNA mutations, while using 'clone' for cells that inherit both lineage markers and transformative phenotypic states such as metaplastic or dysplastic features. This framework helps determine whether disease arose through multiple independent transformation events (polyclonal) or through inheritance from a single transformed ancestor (monoclonal) (Supp. Fig. S7) .

As the reviewer suggests, we have also added a figure to distinguish these scenarios (**Response Fig. 1**).

Response Fig. 1, Supp. Fig. S7 Polyclonal versus monoclonal models of disease development tracked by lineage markers.

Schematic illustrating how lineage markers (represented by different colors) reveal distinct modes of tissue transformation. Polyclonal model where multiple cells independently acquire a disease phenotype (e.g., Barrett's esophagus). Different lineages each contain cells that have independently transformed to the disease state, resulting in phenotypic uniformity despite diverse origins. Monoclonal model where a single cell acquires the disease phenotype and transmits it to all descendants (e.g., dysplasia). All disease cells share both the lineage marker and the inherited pathological phenotype. Normal cells that did not transform retain their original phenotypes.

2. Given that the authors only analyzed dysplastic BE from one patient, they should soften their conclusions derived from this single sample. These include the monoclonal origin of dysplastic BE's (there is always a chance that microscopic dysplastic lesions were not adequately captured in the biopsy), and the inference that APC inactivation promotes dysplastic BE progression.

We appreciate the reviewer's important point about the need to temper our conclusions given that our detailed analysis of dysplastic BE comes from a limited number of samples. We would like to clarify that we found evidence of a clonal expansion in two dysplastic samples in our study: Patient 6 with high-grade dysplasia (analyzed in detail in Fig. 4) and a second patient with low-grade dysplasia (Fig. S18d). Nevertheless, we agree that our findings should be presented as observations from our study rather than universal conclusions about all dysplastic BE. We have revised the manuscript to soften our language in the following places:

1. Abstract statement:
 - a. *In contrast, the precancerous dysplastic lesions showed expansion from a single molecularly aberrant Barrett's esophagus clone.*
2. Results:
 - a. *Strikingly, the dysplastic tissue in this patient originated from the expansion of a single cell containing mtDNA mutation 15153G>A (Fig. 4b, Fig. S18a-c, Fig. S19), a finding further supported by concordant nuclear SNV patterns observed in single-cell RNA-seq (Fig. S9c).*
 - b. *This result raises the possibility that the dysplastic cells in this patient may have been driven by a mechanism recently discovered in premalignant colon adenomas, in which APC-mutant stem cells secrete NOTUM, and thereby shut down and outcompete neighboring wild-type ones^{47,48}.*
3. Discussion:
 - a. *We observed evidence of clonal expansion in two dysplastic samples analyzed. In the case we examined in detail, this expansion could be traced from BE cells through a WNT-dysregulated mechanism. The generalizability of both the BE-to-dysplasia lineage relationship and the specific molecular pathways involved requires investigation in larger cohorts.*

Reviewer #2 (Remarks to the Author):

Reviewer #3 (Remarks to the Author):

The authors have substantially modified their study and introduced significant changes to the manuscript including new analysis that support their model conclusions. Importantly the authors have clarified central message of this study. This version of manuscript provides important evidence in support of polyclonal nature of non-dysplastic BE and clonal nature of dysplastic BE. Secondly, their scRNA-seq data from over 100K cells are a great additional to the field, further strengthened by the spatial validation of these results. In general, the authors have done a good job modifying this manuscript to retain information that are of importance to the field. I believe that the investigation of the cellular composition of BE and the clonal-polyclonal dichotomy between BE with dysplasia and nondysplastic BE should be the central messages of this study. In the case relatedness of tissues of the upper gastro-intestinal tract, I still have few concerns that stem from potential issues with the procedure for the discovery of somatic mitochondrial variants. At this point I would like to raise two points:

1) In the response to my earlier comments, the authors have provided commentary on the mechanistic of empirical FPR calculation using binning methods every 20 coverage counts. However, how do authors calculate FDR for coverage bins which do not have any cells in the control condition. E.g. on figure S6, variant 11891G>A does not have any cells in the control condition with coverage >200 but there are many cells with this coverage in the BE samples. How was FDR calculated for those bins?

Thanks to the reviewer for the thoughtful question. Yes, if there are no cells in the control condition in a given coverage bin for a sample, then we can not calculate the coverage-specific empirical FDR. In this case, we would default back to the overall FDR, which uses the Benjamini-Hochberg procedure under the ZIBB model to get an overall, non-coverage specific FDR. The overall FDR is controlled at 10%. The coverage-based empirical FDR is meant to improve upon this estimate, when control data is available. Within the ZIBB code, the FDR is a settable parameter available to the user to change as they need. To address the concern that our FDR procedure might be too relaxed, we have implemented adjustable FDR thresholds and provide

comprehensive analysis at multiple stringency levels for the samples in Fig. 1d. The table below demonstrates how our ZIBB model performs across different FDR cutoffs (10%, 1%, and 0.1%) for a representative subset of samples:

Patient Number	Variant	Sample	Cell type Order of Numbers	10% sig cells	1% sig cells	0.1% sigcells	10% FalsePositives	1% FalsePositives	0.1% FalsePositives
3	947T>C	A187_be	BE, SQ	573,12	531,10	512,9	29.3,1.57	28.9,1.57	26.2,1.29
3	947T>C	A187_es	SQ	37	37	20	0	0	0
3	947T>C	A187_st	GC	0	0	0	0	0	0
3	2063G>A	A187_be	BE, SQ	67,1	62,1	59,1	15.0,1.011	10.7,1.011	8.58,1.011
3	2063G>A	A187_es	SQ	0	0	0	0	0	0
3	2063G>A	A187_st	GC	4	4	3	0	0	0
8	3047G>A	A193_be1	BE	20	18	15	0	0	0
8	3047G>A	A193_be2	BE	0	0	0	0	0	0
8	3047G>A	A193_es	SQ	125	91	81	0	0	0
8	3047G>A	A193_st	GC	2	2	2	0.001	0.001	0.001

Table 1: Impact of FDR stringency on mitochondrial variant detection and false positive estimation. Representative analysis showing how different FDR thresholds (10%, 1%, and 0.1%) affect both the number of significant cells detected and the estimated false positive counts for selected mitochondrial variants across patients A187 and A193. Each row represents a specific variant-sample combination, with tissue types indicated (BE = Barrett's esophagus, es = esophagus, st = stomach, Normal = control cells).

This table is organized by the different samples from each patient. Multiple numbers correspond to different cell types found in that biopsy for which the order is given by the final column. For example, variant 947T>C in Patient 3 Barrett's esophagus sample shows 573, 531, and 512 significant Barrett's cells at 10%, 1%, and 0.1% FDR respectively, with corresponding decreases in false positive estimates from 29.3 to 26.2. The 2063G>A variant in Patient 3's gastric cardia cell type, which has low cell count, maintains 4, 4, and 3 significant cells across the 10%, 1%, and 0.1% thresholds respectively, indicating the robustness of this biological signal.

This pattern holds consistently across all tested variants and samples, indicating that our statistical framework responds appropriately to different stringency requirements. Even at the most permissive 10% FDR threshold, false positive estimates remain low relative to significant cell counts for variants with strong biological signal. In our initial handling of the data, we did not include any variants where the number of false positives predicted was greater than the signal. Our default 10% FDR represents a standard threshold widely used in genomics applications, but the adjustable implementation allows researchers to apply more conservative cutoffs when warranted. The complete analysis across all samples and FDR thresholds is available in our GitHub repository.

a. it would be helpful if such plots were readily available for each sample either as a single supplementary pdf or at least as fixed link supplementary data (not google drive folder).

These files have been combined by patient and added to the Github for easy viewing.

The link is at https://github.com/jiazhen-rong/scmtVT/tree/master/results/diagnostic_plots_all_samples.

2) Secondly, the choice of empirical FDR is commendably, however, my concern is that the procedure for FDR calculation might be too relaxed. As a result, I have concerns whether some of the mutations identified in specific samples are not true discoveries but false positive results not accounted for by the authors statistical procedure.

For example, mutations 2063G>A in stomach of patient 3 is supported only by 3 or 4 cells (figure 1d), however closer examination of the underlying data

(https://drive.google.com/file/d/1LQlqsJtd5YZs7bGjoZfA2_IDzXeeLbUb/view) demonstrate that the significant cells have low total count of alleles indicating potential false positive result not detected by the authors' FDR calculation. This stands in stark contrast with the data for the same allele in the BE samples (https://drive.google.com/file/d/1GMQfJe95Q9dv2b4I6vMq9IU1Cs_0RgJj/view?usp=drive_link). Here, the mutated alleles are plentiful and they are supported by cells with both high and low total count of reads.

To reiterate, I do not think the statistical procedures used in the ZIBB model are incorrect. My concern is that the chosen FDR calculation procedure is not stringent enough to support authors conclusions about shared lineage between tissue types when the number of cells supporting this conclusion is limited.

We thank the reviewer for their careful examination of variant 2063G>A. Indeed, for this example, the number of putative carriers in stomach is only 4. While on the diagnostic plot it looks like the coverage of these putative carrier cells is low, that is because of the large range of the x axis. The coverage is, in fact, quite high: 154, 316, 160, and 300. The box plot below compares the coverage of these putative carriers to that in the Barrett's esophagus samples, and we see that they are comparable (**Response Fig. 2, left**). Also, within the stomach sample, the coverage of the 4 putative carriers at this site is also comparable to that of the non-carriers, which indicates that they are not false positives due to “low-quality” cells (**Response Fig. 2, right**). Within these 4 cells, the alternative allele counts are all very high (the lowest is 10). In comparison, the highest allele count in normal cells is <10. This indicates that we can not explain this high alternative allele count based on contamination or dispersion. This gives us confidence that these 4 cells are true carriers.

Response Fig. 2. Right plot: Coverage of 2063G>A in the positive cells in the Barrett's esophagus versus stomach samples. Statistical comparison shows no significant difference in coverage ($p=0.36$, Wilcoxon test) or allele frequency ($p=0.59$, Wilcoxon test) between positive cells in the stomach versus BE samples. Left plot: Coverage of cells positive (carrier) versus negative (non-carrier) in the stomach sample. Statistical comparison shows no significant difference in coverage between carrier and non-carrier ($p=0.211$, Wilcoxon test)

a. As a result, it would be important that at minimum the authors introduce additional discussion points to account for the fact that some of the shared mutations might be artefacts of their analysis. Alternatively, the authors should investigate if the conclusions stand if more stringent FDR procedures are applied.

We agree that there may be examples where artifacts could slip through filtering considerations. We have added a line to the discussion emphasizing the importance of inspecting the data and considering sources of error in using these methods. Here is the new sentence in the discussion section:

While the orthogonal SNV validation and ZIBB provide confidence for lineage assignments, we acknowledge that artifacts are possible, particularly in the context of small numbers of cells containing a variant, thus it is important to carefully inspect the underlying data (Fig. S5, S6, S10, and scmtVT Github) and consider more stringent filtering.

b. Also, the abstract still contains a statement “Our analysis unexpectedly revealed that the squamous esophagus and gastric cardia contained cells belonging to common lineages that also included transitional basal cells; both esophageal and gastric tissues were also related to Barrett’s esophagus”. In light of my concern, this statement cannot be easily supported by the results presented in this study and it should not treat it as the main conclusion of their analysis that should be included in the abstract.

We believe this is still an important finding, but have softened the claims using the term “evidence for” to present this result in the abstract. Here is the new wording:

Our analysis unexpectedly revealed evidence for lineages spanning squamous esophagus, gastric cardia, and transitional basal cells at the tissue junction.

Minor point:

In figure S7, the authors introduced heatmaps “Ordered by Biopsy”. I believe that unlike figures to the right of these heatmaps, all cells were plotted. As a result, there is a discrepancy between the panels - e.g. mutations 2063G>A does not exist in blue (esophagus) Sample on the left panel but many cells in the same sample have it at low AF on the right panel.

This is correct that the “Ordered by Biopsy” plots only show the cells that contain a variant. We have updated the figure legend to make this information available to the reader.

Reviewer #3 (Remarks on code availability):

The code is accessible, well annotated and appears to be reproducible.

Reviewer #5 (Remarks to the Author): Expert in single-cell RNA-seq and clonal lineage tracing; replaces Reviewer #4

This revised manuscript by Gier et al focuses on the study of Barrett’s esophagus, a premalignant condition of esophageal adenocarcinoma. The major claims of the article are that:

1. Squamous esophagus and gastric cardia contain cells belonging to common lineages that included transitional basal cells
2. Normal esophageal and gastric tissues contain clones that are related to Barrett’s dysplastic cells (which suggests a common ancestral origin, or possible sources from both organs as the cell of origin)
3. Barrett’s esophagus in its early state contains multiple clones (possibly multiple origins)
4. More dysplastic cells contain a single molecularly aberrant clone (derived from one of the BE clones/subclones in a classic clonal evolution process)

Evidence to support these claims mostly comes from a series of biopsies from various patients that were analyzed through single-cell RNAseq and a modified version of MAESTER to enrich and profile mitochondrial mutations for lineage tracing. Methods seem to rely on state-of-the-art, validated techniques, for the most part (the ZIBB model is a nice addition), and the quality controls and stringency measures seem to be appropriate and well demonstrated in supplementary figures. I could also find and run the ZIBB code from github.

Fig 1 presents the general evidence for claims 1-2. I found support figures S8 (SNV validation) and S9 (doublet/cell-state testing) very convincing. Fig 2 presents the evidence that Barrett’s contains multiple clones and each clone appears to be multilineage, being in basically every cell type (claim 3). Fig 3 presents spatial transcriptomics of Barrett’s, which validates some new BE cell types, which is really good. Fig 4 presents the

evidence of claim 4, that dysplastic cells are basically a single clone derived from one of the BE clones (subclones?). WGS indicates the likely mutational process, and the evidence of transcriptionally similar cells in at least some esophageal adenocarcinomas (main supporting evidence that these dysplasias identified would go on to become EAC).

Regarding the points raised by other reviewers, Reviewer 1 technical comments are valid, and highlight some of the known limitations of mt-based tracing. First, mitochondrial tracing is rarely based on fully homoplasmic mutations... usually there is some heteroplasmy, and the distributions of VAFs are hard to interpret due to limitations (copy numbers, dropouts, etc.). I think the reply by the authors is fair, confounders are excluded as much as possible, and they even went to the extent of validating variant detection and calls using the ZIBB model and performing SNV-based validation (Fig S8). I would be satisfied with this answer.

Regarding Reviewer 1 conceptual comments, I think that it is also valid to argue that all these cells have an ancestral clone. The whole animal is coming from a single ancestral clone if we consider the zygote. With our current knowledge of how long it takes for mitochondrial variants to arise and stabilize at high heteroplasmy, and the fact that these clones differentiate to most BE cell types, it is reasonable to argue that BE may have independent origins, even if (as the Reviewer points out) the killer experiment is to show normal non-BE cells with the same mutations in order to exclude the fact that these mt variants arose within the BE-tissue. In any case, I believe there is significant novelty in showing that BE is a polyclonal ecosystem. Meaning, at the very least, the authors are proving the existence of multiple BE “stem cells” and their fate behaviors, which is super interesting and novel. The manuscript could better highlight in the abstract the finding and validation of novel BE cell types (validation both from the clonal perspective, since they share mt variants with other BE types, and from the seqFISH experiment). Finally, I do agree with their MINOR comment 5, regarding some overstatements.

Reviewer 3 point 1 goes very much along the lines of Reviewer 1 at least regarding concepts. I think the authors have done a good job at toning down their polyclonal origin model in favor of focusing on a polyclonal state model. I also want to highlight that the replies regarding QC of variants (Fig S5) and doublets (Fig S9) are easy to follow and understand, and I agree generally with their interpretation.

I also want to comment on the Reviewer 4 point 2 regarding the lack of sufficient evidence for a single progenitor cell for BE. This is a classical problem that emerges with mt-based tracing, since we cannot fully time the emergence of the mitochondrial variants. In one patient, the stabilized variant may be late and fate-restricted. In the next patient, the variant may be even representative of clonal origins pre-BE. This is why it is so important in this kind of study to present these data cautiously without overinterpreting. But this is not to say the study is not novel (as reviewer 1 so adamantly states). I think it pushes the boundaries of what has been done before, and it provides guidance for future endeavors trying to solve these questions (with mt mutations or otherwise).

In Figure 4b/c, the tree connection between BE and Dysplasia cells is very cool, but really hard for me to see. It's too small, and there's too few cells in the BE lineage block of the heatmap. I suggest downsampling the Dysp cells for plotting the heatmap (they're all mostly the same anyways!).

In summary, from my assessment at the current review stage, considering the manuscript in its current state, I believe it would still benefit from some further text adjustments, but no further experiments or analysis would be necessary.

We thank the reviewer for their thorough and supportive assessment of our manuscript. We appreciate their recognition of our methodological rigor and the novelty of our findings regarding the polyclonal nature of Barrett's esophagus. Based on these suggestions, we have implemented the following changes:

1. Following this reviewer's and Reviewer 1's suggestions, we have further refined our language throughout the manuscript to ensure our claims are appropriately measured. This was done in the abstract, results and discussion as outlined above.

2. We have revised the abstract to better highlight our discovery and validation of novel BE cell types, which were confirmed both through clonal analysis showing they share mitochondrial variants with other BE cell types and through spatial transcriptomics validation. Here is the new sentence in the abstract:

Our analysis identified previously uncharacterized Barrett's esophagus cell types including tuft, ciliated, and BEST4+ cells, validated through both lineage relationships and spatial transcriptomics.

3. We have improved the visualization in Fig. 4b/c by downsampling the dysplastic cells in the heatmap to 200 representative cells (maintaining the full clonal diversity) and enlarging the phylogenetic tree panel. This makes the lineage relationships between BE and dysplastic cells much clearer. This was an excellent suggestion!

Response Fig. 3: Panels b and c of Figure 4. Updated heatmaps with suggested subsampling and enlargement of the phylogenetic tree panel for clearer visualization.

Reviewer #5 (Remarks on code availability):

I think the code and data should be deposited somewhere more definitive with a proper DOI before final acceptance.

The data is all deposited into GEO. The ZIBB pipeline is on Github (scmtVT) along with PDFs of the diagnostic plots and tables of the new FDR thresholding analysis for the samples in the manuscript. The code for generating the figures has now been deposited into Zenodo at [10.5281/zenodo.16997177](https://zenodo.org/record/16997177).